# RETI-DIFF: ILLUMINATION DEGRADATION IMAGE RESTORATION WITH RETINEX-BASED LATENT DIFFUSION MODEL

**Chunming He**[1,*], **Chengyu Fang**[2,*,†] , **Yulun Zhang**[3,†] , **Longxiang Tang**[2] ,
**Jinfa Huang**[4] , **Kai Li**[5] , **Zhenhua Guo**[6] , **Xiu Li**[2] , and **Sina Farsiu**[1,†]
[1]Duke University, [2]Tsinghua University, [3]Shanghai Jiao Tong University,
[4]Peking University, [5]Meta, [6]Tianyi Traffic Technology,

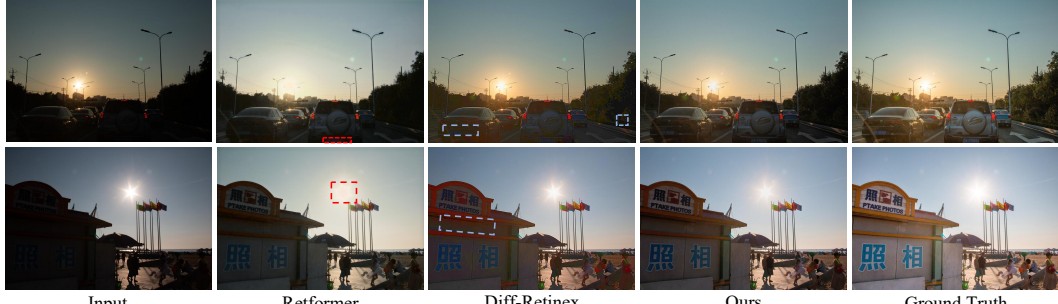

Figure 1: Results of Retinex-based cutting-edge image restoration methods, where our Reti-Diff can better highlight details and correct color distortions. The dashed boxes indicate failure cases or artifacts produced by existing methods, which can be properly addressed by our approach.

## ABSTRACT

Illumination degradation image restoration (IDIR) techniques aim to improve the visibility of degraded images and mitigate the adverse effects of deteriorated illumination. Among these algorithms, diffusion-based models (DM) have shown promising performance but are often burdened by heavy computational demands and pixel misalignment issues when predicting the image-level distribution. To tackle these problems, we propose to leverage DM within a compact latent space to generate concise guidance priors and introduce a novel solution called Reti-Diff for the IDIR task. Specifically, Reti-Diff comprises two significant components: the Retinex-based latent DM (RLDM) and the Retinex-guided transformer (RGformer). RLDM is designed to acquire Retinex knowledge, extracting reflectance and illumination priors to facilitate detailed reconstruction and illumination correction. RGformer subsequently utilizes these compact priors to guide the decomposition of image features into their respective reflectance and illumination components. Following this, RGformer further enhances and consolidates these decomposed features, resulting in the production of refined images with consistent content and robustness to handle complex degradation scenarios. Extensive experiments demonstrate that Reti-Diff outperforms existing methods on three IDIR tasks, as well as downstream applications. The source code is available at https://github.com/ChunmingHe/Reti-Diff.

## 1 INTRODUCTION

Illumination degradation image restoration (IDIR) seeks to enhance the visibility and contrast of degraded images while mitigating the adverse effects of deteriorated illumination, *e.g.*, indefinite noise and variable color deviation. IDIR has been investigated in various domains, including low-light image enhancement (Cai et al., 2023), underwater image enhancement (Guo et al., 2023), and backlit image enhancement (Liang et al., 2023). By addressing illumination degradation, the

---

*Equal Contribution, † Corresponding Author, Contact: chunming.he@duke.edu.

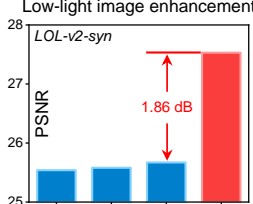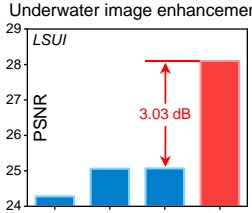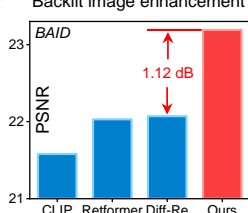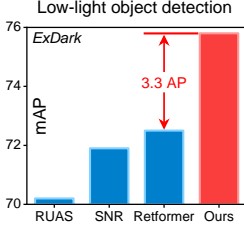

Figure 2: Our Reti-Diff achieves a leading place in three IDIR tasks and the low-light object detection task, and outperforms the corresponding cutting-edge techniques on these tasks, where CLIP and Diff-Re are short for CLIP-LIT (Liang et al., 2023), Diff-Retinex (Yi et al., 2023).

enhanced images are expected to exhibit improved visual quality, making them more suitable for decision-making or subsequent tasks like nighttime object detection and segmentation.

Traditional IDIR approaches (Fu et al., 2016a; Ueng & Scharf, 1995) primarily rely on manually crafted enhancement techniques with limited generalization capabilities. Leveraging the robust feature extraction capabilities of convolutional neural networks and transformers, a series of deep learning-based methods (Cai et al., 2023; Jiang et al., 2021) have been proposed and have achieved remarkable success in the IDIR domain. However, as depicted in Figs. 1 and 2, they still face challenges in complex illumination degradation scenarios due to their constrained restoration capacity.

To overcome this, deep generative models, like generative adversarial networks (He et al., 2023a), have gained popularity for their generative abilities. Recently, the diffusion model (DM) (Yi et al., 2023) has been introduced to the IDIR field for high-quality image restoration. However, existing DM-based methods, *e.g.*, Diff-Retinex (Yi et al., 2023) and GSAD (Jinhui et al., 2023), apply DM directly to image-level generation, leading to two main challenges: **(1)** These methods incur high computational costs, as predicting the image-level distribution requires a large number of inference steps. **(2)** The enhanced results may exhibit pixel misalignment with the original clean image in terms of restored details and local consistency. For example, as shown in Fig. 1, Diff-Retinex fails to recover the car's details in the top row and introduces severe artifacts in the bottom row.

To address the above challenges, we introduce a latent diffusion model (LDM) to solve the IDIR problem. The computational burden is reduced by applying DM in the low-dimensional, compact latent space. In addition, by integrating LDM with transformers, we prevent pixel misalignment in generated images (see Fig. 1), a common issue in deep generative models. Unlike existing LDM-based methods that rely solely on priors extracted from the RGB domain, our method, tailored to the specific characteristics of IDIR tasks, empowers LDMs to extract Retinex information from both the reflectance and illumination domains. This adaptation allows our method to generate high-fidelity Retinex priors directly from low-quality input images. The compact priors preserve high-quality information while minimizing the impact of degradation. Thus, our method simultaneously enhances image details using the reflectance prior and corrects color distortions with the illumination prior, resulting in visually appealing images with favorable downstream tasks.

With this inspiration, we present Reti-Diff, the first LDM-based solution to tackle the IDIR problem. Reti-Diff, depicted in Fig. 3, consists of two primary components: the Retinex-based LDM (RLDM) and the Retinex-guided transformer (RGformer). Initially, RLDM is employed to generate Retinex priors, which are then integrated into RGformer to produce visually appealing results. To ensure the generation of high-quality priors, we propose a two-phase training approach, wherein Reti-Diff undergoes initial pretraining followed by subsequent RLDM optimization. **In phase I**, we introduce a Retinex prior extraction (RPE) module to compress the ground-truth image into the highly compact Retinex priors, namely the reflectance prior and the illumination prior. These priors are then sent to RGformer to guide feature decomposition and the generation of reflectance and illumination features. Afterward, RGformer employs the Retinex-guided multi-head cross attention (RG-MCA) and dynamic feature aggregation (DFA) module to refine and aggregate the decomposed features, ultimately producing enhanced images with coherent content and ensuring robustness and generalization in extreme degradation scenarios. **In phase II**, we train RLDM in reflectance and illumination domains to estimate Retinex priors from the low-quality image, with the constraint of consistency with those extracted by RPE from the ground-truth image. Therefore, the extracted Retinex priors can guide the RGformer in detail enhancement and illumination correction, resulting in visually appealing results with favorable downstream performance.

Our contributions are summarized as follows:

- We propose a novel DM-based framework, Reti-Diff, for the IDIR task. To the best of our knowledge, this is the first practice of the latent diffusion model to tackle the IDIR problem.

- We propose to let RLDM learn Retinex knowledge and generate high-quality reflectance and illumination priors from the low-quality input, which serve as critical guidance in detail enhancement and illumination correction and can be integrated with various methods.

- We propose RGformer, which integrates extracted Retinex priors to decompose features into reflectance and illumination components. Subsequently, RG-MCA and DFA are employed to refine and aggregate these decomposed features, ensuring robustness and generalization in complex illumination degradation scenarios.

- Extensive experiments on four IDIR tasks verify our superiority, efficiency, and generalizability to existing methods in terms of image quality and favorability in downstream applications, including low-light object detection and image segmentation.

## 2 RELATED WORK

**Illumination Degradation Image Restoration.** Early IDIR methods mainly include three approaches: histogram equalization (HE) (Cheng & Shi, 2004), gamma correction (GC) (Huang et al., 2012), and Retinex theory (Land, 1977). HE-based and GC-based methods focused on directly amplifying the low contrast regions but ignore illumination factors. Retinex-based variants (Fu et al., 2016b; Li et al., 2018) proposed priors to constrain the solution space for reflectance and illumination maps. However, these methods still rely on hand-crafted priors, limiting their generalization ability. With the development of deep learning, methods based on CNNs and transformers (Peng et al., 2025; He et al., 2023a; Jin et al., 2022; 2023; Pu et al., 2024; Li et al., 2020) have succeeded in IDIR. For instance, DIE (Wang et al., 2019) integrated Retinex cues into a learning-based structure, presenting a one-stage Retinex-based solution for color correction. To enhance generative capacity, Diff-Retinex (Yi et al., 2023) and GSAD (Jinhui et al., 2023) introduced DM to the IDIR field by directly applying it to image-level generation. However, they entail significant computational costs and may lead to pixel misalignment with the original input, particularly concerning restored image details and local consistency.

**Diffusion Models.** Diffusion models (DMs) have verified great success in density estimation (Kingma et al., 2021; Lin et al., 2024) and data generation (He et al., 2024a; Zhu et al., 2024b;a). Such a probabilistic generative model adopts a parameterized Markov chain to optimize the lower variational bound on the likelihood function, enabling them to generate target distributions with greater accuracy. Recently, DMs have been introduced to solve the IDIR problem (Yi et al., 2023; Jinhui et al., 2023; Jin et al., 2024a;b; He et al., 2023b). However, when directly applied to image-level generation, these methods bring computational burdens and pixel misalignment. To overcome this, we employ LDM to estimate priors within a low-dimensional latent space and then integrate these priors into the transformer-based framework, addressing the above problems. Besides, unlike existing LDM-based methods (Xia et al., 2023; Chen et al., 2023) that solely rely on priors extracted from the RGB domain, our method, tailored to the IDIR task, empowers LDMs to extract Retinex information from both the reflectance and illumination domains. This adaptation allows our method to generate high-fidelity compact Retinex priors directly from low-quality input images but avoid the impact of degradation. By doing so, this novel approach enables us to simultaneously enhance image details using the reflectance prior and correct color distortions with the illumination prior, resulting in visually appealing results with favorable downstream tasks.

## 3 METHODOLOGY

In this paper, we propose Reti-Diff, the pioneering method based on Latent Diffusion Models (LDM) for IDIR tasks. Reti-Diff is specifically tailored to address the challenges inherent in IDIR tasks by leveraging high-quality Retinex priors extracted from both the illumination and reflectance domains to guide the restoration process. This innovative approach utilizes the extracted Retinex prior representation as dynamic modulation parameters, facilitating simultaneous enhancement of restoration details through the reflectance prior and correction of color distortion via the illumination prior. This ensures the generation of visually compelling results while positively impacting downstream tasks.

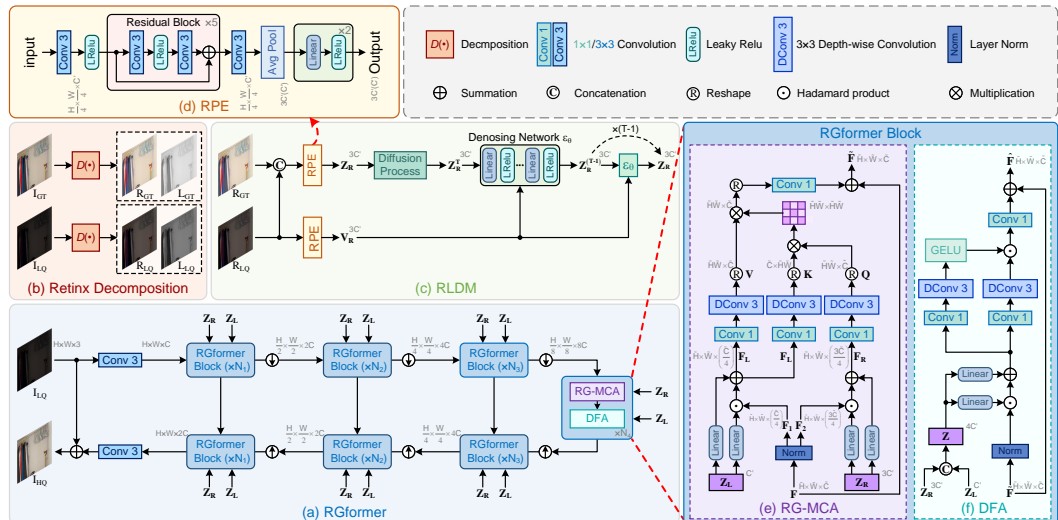

Figure 3: Framework of Reti-Diff. In Phase I, we pretrain Reti-Diff with RGformer and RPE to ensure the robust learning of RLDM and then optimize RLDM to generate high-quality Retinex priors in Phase II, which guide RGformer in detail enhancement and illumination correction. In (a), we omit the auxiliary decoder $D_a(\cdot)$ for simplicity. In panel (c), we illustrate the use of RLDM to extract the reflectance prior; the illumination prior can be extracted similarly. Zoom in for clarity.

As shown in Fig. 3, our Reti-Diff comprises two parts: the Retinex-guided transformer (RGformer) and the Retinex-based latent diffusion model (RLDM). To ensure the generation of high-quality priors, Reti-Diff undergoes a two-phase training strategy, involving the initial pretraining of Reti-Diff and the subsequent optimization of RLDM. In this section, we provide an in-depth explanation of the two-phase training approach and elucidate the entire restoration process.

## 3.1 PRETRAIN RETI-DIFF

We first pretrain Reti-Diff to encode the ground truth image into compact priors with Retinex prior extraction (RPE) module and use the extracted Retinex priors to guide RGformer for restoration.

**Retinex prior extraction module.** Given the low-quality (LQ) image $\mathbf{I}_{LQ} \in \mathbb{R}^{H \times W \times 3}$ and its corresponding ground truth $\mathbf{I}_{GT} \in \mathbb{R}^{H \times W \times 3}$, we initially decompose them into the reflectance image $\mathbf{R} \in \mathbb{R}^{H \times W \times 3}$ and the illumination map $\mathbf{L} \in \mathbb{R}^{H \times W}$ according to Retinex theory:

$$\mathbf{I}_{LQ} = \mathbf{R}_{LQ} \odot \mathbf{L}_{LQ}, \mathbf{I}_{GT} = \mathbf{R}_{GT} \odot \mathbf{L}_{GT}, \tag{1}$$

where $\odot$ is Hadamard product. Following URetinex (Wu et al., 2022), we use a pretrained decomposing network $D(\cdot)$ to decompose $\mathbf{I}_{LQ}$ and $\mathbf{I}_{GT}$, comprising three Conv+LeakyReLU layers and a Conv+ReLU layer. Then we concatenate the corresponding components of ground truth and LQ image and use the RPE module RPE($\cdot$) to encode them into Retinex priors $\mathbf{Z_R} \in \mathbb{R}^{3C'}$, $\mathbf{Z_L} \in \mathbb{R}^{C'}$:

$$\mathbf{Z_R} = \text{RPE}(\text{down}(\text{conca}(\mathbf{R}_{GT}, \mathbf{R}_{LQ}))), \quad \mathbf{Z_L} = \text{RPE}(\text{down}(\text{conca}(\mathbf{L}_{GT}, \mathbf{L}_{LQ}))), \tag{2}$$

where conca($\cdot$) denotes concatenation and down($\cdot$) represents downsampling that is operated by PixelUnshuffle. The Retinex priors, $\mathbf{Z_R}$ and $\mathbf{Z_L}$, are then fed into RGformer to serve as dynamic modulation parameters for detail restoration and color correction.

**Retinex-guided transformer.** RGformer mainly consists of two parts in each block, *i.e.*, Retinex-guided multi-head cross attention (RG-MCA) and dynamic feature aggregation (DFA) module. In RG-MCA, we first split the input feature $\mathbf{F} \in \mathbb{R}^{\tilde{H} \times \tilde{W} \times \tilde{C}}$ into two parts $\mathbf{F}_1 \in \mathbb{R}^{\tilde{H} \times \tilde{W} \times (3\tilde{C}/4)}$ and $\mathbf{F}_2 \in \mathbb{R}^{\tilde{H} \times \tilde{W} \times (\tilde{C}/4)}$ along the channel dimension. Afterwards, we integrated $\mathbf{Z_R}$ and $\mathbf{Z_L}$ as the corresponding dynamic modulation parameters to generate reflectance-guided feature $\mathbf{F_R} \in \mathbb{R}^{\tilde{H} \times \tilde{W} \times (3\tilde{C}/4)}$ and illumination-guided feature $\mathbf{F_L} \in \mathbb{R}^{\tilde{H} \times \tilde{W} \times (\tilde{C}/4)}$:

$$\mathbf{F_R} = \text{Li}_1(\mathbf{Z_R}) \odot \text{Norm}(\mathbf{F}_1) + \text{Li}_2(\mathbf{Z_R}), \quad \mathbf{F_L} = \text{Li}_1(\mathbf{Z_L}) \odot \text{Norm}(\mathbf{F}_2) + \text{Li}_2(\mathbf{Z_L}), \tag{3}$$

where Norm($\cdot$) is layer normalization. Li($\cdot$) means linear layer. Afterward, we aggregate global spatial information by projecting $\mathbf{F_R}$ into query $\mathbf{Q} = \mathbf{W}_Q \mathbf{F_R}$ and key $\mathbf{K} = \mathbf{W}_K \mathbf{F_L}$ and transforming $\mathbf{F_L}$ into value $\mathbf{V} = \mathbf{W}_V \mathbf{F_L}$, where $\mathbf{W}$ is the combination of a $1 \times 1$ point-wise convolution and a

$3 \times 3$ depth-wise convolution. We then perform cross-attention and get the output feature $\tilde{\mathbf{F}}$:

$$\tilde{\mathbf{F}} = \mathbf{F} + \text{SoftMax}\left(\mathbf{Q}\mathbf{K}^T/\sqrt{\tilde{C}}\right) \cdot \mathbf{V}. \tag{4}$$

By doing so, RG-MCA introduces explicit guidance to fully exploit Retinex knowledge at the feature level and use cross attention mechanism to implicitly model the Retinex theory and refine the decomposed features, which helps to restore missing details and correct color distortion.

Then we employ DFA for local feature aggregation. Apart from the $1 \times 1$ Conv and $3 \times 3$ depth-wise Conv for information fusion, DFA adopts GELU, termed GELU($\cdot$), to ensure the flexibility of aggregation (He et al., 2023c). Thus, given $\tilde{\mathbf{F}}$ and $\mathbf{Z}$, where $\mathbf{Z} = \text{conca}(\mathbf{Z_R}, \mathbf{Z_L})$, the output $\hat{\mathbf{F}}$ is

$$\hat{\mathbf{F}} = \tilde{\mathbf{F}} + \text{GELU}(\mathbf{W}_1\mathbf{F}') \odot \mathbf{W}_2\mathbf{F}', \quad \mathbf{F}' = \text{Li}_1(\mathbf{Z}) \odot \text{Norm}(\tilde{\mathbf{F}}) + \text{Li}_2(\mathbf{Z}). \tag{5}$$

**Optimization.** Having gotten the enhanced result $\mathbf{I}_{HQ}$, we propose a reconstruction loss with $L_1$ norm $\|\cdot\|_1$ to jointly train RPE and RGformer, which can facilitate the extraction of Retinex priors:

$$L_{Rec} = \|\mathbf{I}_{GT} - \mathbf{I}_{HQ}\|_1. \tag{6}$$

To ensure that the separated features within RG-MCA capture reflectance and illumination knowledge, we use an auxiliary decoder $D_a(\cdot)$ with the same structure as that in (Locatello et al., 2020). $D_a(\cdot)$ takes $\tilde{\mathbf{F}}$ as input and outputs the reconstructed reflectance image $\mathbf{R}_{Re}$ and illumination map $\mathbf{L}_{Re}$. For efficiency, we only apply $D_a(\cdot)$ for the first transformer block in encoder to get $\mathbf{R}^I_{Re}$ and $\mathbf{L}^I_{Re}$ and for the last block in decoder to get $\mathbf{R}^L_{Re}$ and $\mathbf{L}^L_{Re}$. $D_a(\cdot)$ is supervised by a Retinex loss:

$$L_R = \|\mathbf{R}_{LQ} - \mathbf{R}^I_{Re}\|_1 + \|\mathbf{L}_{LQ} - \mathbf{L}^I_{Re}\|_1 + \|\mathbf{R}_{GT} - \mathbf{R}^L_{Re}\|_1 + \|\mathbf{L}_{GT} - \mathbf{L}^L_{Re}\|_1. \tag{7}$$

By constraining the input and output ports, Eq. (7) ensures the preservation of essential Retinex information throughout the network. This integration not only facilitates the incorporation of Retinex theory into the split features but also enhances the overall restoration capability.

In Phase I, the final loss $L_{P1}$ is formulated with the assistance of a hyperparameter $\lambda_1$ ($\lambda_1 = 1$):

$$L_{P1} = L_{Rec} + \lambda_1 L_R. \tag{8}$$

## 3.2 RETINEX-BASED LATENT DIFFUSION MODEL

In Phase II, we train the RLDM to predict Retinex priors from the low-quality input, which are expected to be consistent with that extracted by RPE from the ground-truth image. Unlike conventional LDMs trained on the RGB domain, we introduce two RLDMs with a Siamese structure and train them on distinct domains: the reflectance domain and the illumination domain. This approach, grounded in Retinex theory, equips our RLDM to generate a more generative reflectance prior $\hat{\mathbf{Z}}_\mathbf{R}$ to enhance image details, and a more harmonized illumination prior $\hat{\mathbf{Z}}_\mathbf{L}$ for color correction. The compact priors retain high-quality information while effectively mitigating the effects of degradation. Note that RLDM is constructed upon the conditional denoising diffusion probabilistic models, with both a forward diffusion process and a reverse denoising process. To simplify, we provide a detailed derivation for $\hat{\mathbf{Z}}_\mathbf{R}$ herein, while that of $\hat{\mathbf{Z}}_\mathbf{L}$ can be found in the appendix.

**Diffusion process.** In the diffusion process, we first use the pretrained RPE to extract the reflectance prior $\mathbf{Z_R}$, which is treated as the starting point of the forward Markov process, *i.e.*, $\mathbf{Z_R} = \mathbf{Z}^0_\mathbf{R}$. We then gradually add Gaussian noise to $\mathbf{Z_R}$ by $T$ iterations and each iteration can be defined as:

$$q\left(\mathbf{Z}^t_\mathbf{R}|\mathbf{Z}^{t-1}_\mathbf{R}\right) = \mathcal{N}\left(\mathbf{Z}^t_\mathbf{R}; \sqrt{1-\beta^t}\mathbf{Z}^{t-1}_\mathbf{R}, \beta^t\mathbf{I}\right), \tag{9}$$

where $t = 1, \cdots, T$. $\mathbf{Z}^t_\mathbf{R}$ denotes the noisy prior at time step $t$, $\beta^t$ is the predefined factor that controls the noise variance, and $\mathcal{N}$ is the Gaussian distribution. Following (Kingma & Welling, 2013), we define $\alpha^t = 1 - \beta^t$ and $\bar{\alpha}^t = \prod_{i=1}^t \alpha^i$, allowing us to simplify Eq. (9) as follows:

$$q\left(\mathbf{Z}^t_\mathbf{R}|\mathbf{Z}^0_\mathbf{R}\right) = \mathcal{N}\left(\mathbf{Z}^t_\mathbf{R}; \sqrt{\bar{\alpha}^t}\mathbf{Z}^0_\mathbf{R}, (1-\bar{\alpha}^t)\mathbf{I}\right). \tag{10}$$

**Reverse process.** In the reverse process, RLDM aims to extract the reflectance prior from pure Gaussian noise. Thus, RLDM samples a Gaussian random noise map $\mathbf{Z}^T_\mathbf{R}$ and then gradually denoise it to run backward from $\mathbf{Z}^T_\mathbf{R}$ to $\mathbf{Z}^0_\mathbf{R}$ with the corresponding mean $\mu^t$ and variance $\sigma^t$:

$$p(\mathbf{Z}^{t-1}_\mathbf{R}|\mathbf{Z}^t_\mathbf{R}, \mathbf{Z}^0_\mathbf{R}) = \mathcal{N}(\mathbf{Z}^{t-1}_\mathbf{R}; \boldsymbol{\mu}^t(\mathbf{Z}^t_\mathbf{R}, \mathbf{Z}^0_\mathbf{R}), (\boldsymbol{\sigma}^t)^2\mathbf{I}), \tag{11}$$

| Methods | Sources | LOL-v1 | | | | LOL-v2-real | | | | LOL-v2-synthetic | | | | SID | | | |
|---|---|---|---|---|---|---|---|---|---|---|---|---|---|---|---|---|---|
| | | PSNR↑ | SSIM↑ | FID↓ | BIQE↓ | PSNR↑ | SSIM↑ | FID↓ | BIQE↓ | PSNR↑ | SSIM↑ | FID↓ | BIQE↓ | PSNR↑ | SSIM↑ | FID↓ | BIQE↓ |
| MIRNet (Zamir et al., 2020) | ECCV20 | 24.14 | 0.835 | 71.16 | 47.75 | 20.02 | 0.820 | 82.25 | 41.18 | 21.94 | 0.876 | 40.18 | 36.29 | 20.84 | 0.605 | 81.37 | 40.63 |
| EnGAN (Jiang et al., 2021) | TIP21 | 17.48 | 0.656 | 153.98 | 35.82 | 18.23 | 0.617 | 173.28 | 51.06 | 16.57 | 0.734 | 93.66 | 45.59 | 17.23 | 0.543 | 77.52 | 33.47 |
| RUAS (Liu et al., 2021) | CVPR21 | 18.23 | 0.723 | 127.60 | 45.17 | 18.27 | 0.723 | 151.62 | 34.73 | 16.55 | 0.652 | 91.60 | 46.38 | 18.44 | 0.581 | 72.18 | 45.02 |
| IPT (Chen et al., 2021) | CVPR21 | 16.27 | 0.504 | 158.83 | 29.35 | 19.80 | 0.813 | 97.24 | 31.17 | 18.30 | 0.811 | 76.79 | 42.15 | 20.53 | 0.618 | 70.58 | 36.71 |
| URetinex (Wu et al., 2022) | CVPR22 | 21.33 | 0.835 | 85.59 | 30.37 | 20.44 | 0.806 | 76.74 | 28.85 | 24.73 | 0.897 | 33.25 | 33.46 | 22.09 | 0.633 | 71.58 | 38.44 |
| UFormer (Wang et al., 2022) | CVPR22 | 16.36 | 0.771 | 166.69 | 41.06 | 18.82 | 0.771 | 164.41 | 40.36 | 19.66 | 0.871 | 58.69 | 39.75 | 18.54 | 0.577 | 100.14 | 42.13 |
| Restormer (Zamir et al., 2022) | CVPR22 | 22.43 | 0.823 | 78.75 | 33.18 | 19.94 | 0.827 | 114.35 | 37.27 | 21.41 | 0.830 | 46.89 | 35.06 | 22.27 | 0.649 | 75.47 | 32.49 |
| SNR-Net (Xu et al., 2022) | CVPR22 | 24.61 | 0.842 | 66.47 | 28.73 | 21.48 | 0.849 | 68.56 | 28.83 | 24.14 | 0.928 | 30.52 | 33.47 | 22.87 | 0.625 | 74.78 | 30.08 |
| SMG (Xu et al., 2023) | CVPR23 | 24.82 | 0.838 | 69.47 | 30.15 | 22.62 | 0.857 | 71.76 | 30.32 | 25.62 | 0.905 | 23.36 | 29.35 | 23.18 | 0.644 | 77.58 | 31.50 |
| PyDiff (Zhou et al., 2023a) | IJCAI23 | 21.15 | 0.857 | 49.47 | 21.13 | — | — | — | — | — | — | — | — | — | — | — | — |
| Retformer (Cai et al., 2023) | ICCV23 | 25.16 | 0.845 | 72.38 | 26.68 | 22.80 | 0.840 | 79.58 | 34.39 | 25.67 | 0.930 | 22.78 | 30.26 | 24.44 | 0.680 | 82.64 | 35.04 |
| Diff-Retinex (Yi et al., 2023) | ICCV23 | 21.98 | 0.852 | 51.33 | 19.62 | 20.17 | 0.826 | 46.67 | 24.18 | 24.30 | 0.921 | 28.74 | 26.35 | 23.62 | 0.665 | 58.93 | 31.17 |
| MRQ (Liu et al., 2023) | ICCV23 | 25.24 | 0.855 | 53.32 | 22.73 | 22.37 | 0.854 | 68.89 | 33.61 | 25.54 | 0.940 | 20.86 | 25.09 | 24.62 | 0.683 | 61.09 | 27.81 |
| IAGC (Wang et al., 2023) | ICCV23 | 24.53 | 0.842 | 59.73 | 25.50 | 22.20 | 0.863 | 70.34 | 31.70 | 25.58 | 0.941 | 21.38 | 30.32 | 24.80 | 0.688 | 63.72 | 29.53 |
| DiffIR (Xia et al., 2023) | ICCV23 | 23.15 | 0.828 | 70.13 | 26.38 | 21.15 | 0.816 | 72.33 | 29.15 | 24.76 | 0.921 | 28.87 | 27.74 | 23.17 | 0.640 | 78.80 | 30.56 |
| CUE (Zheng et al., 2023) | ICCV23 | 21.86 | 0.841 | 69.83 | 27.15 | 21.19 | 0.829 | 67.05 | 28.83 | 24.41 | 0.917 | 31.33 | 33.83 | 23.25 | 0.652 | 77.38 | 28.85 |
| GSAD (Jinhui et al., 2023) | NIPS23 | 23.23 | 0.852 | 51.64 | 19.96 | 20.19 | 0.847 | 46.77 | 28.85 | 24.22 | 0.927 | 19.24 | 25.76 | — | — | — | — |
| AST (Zhou et al., 2024) | CVPR24 | 21.09 | 0.858 | 87.67 | 21.23 | 21.68 | 0.856 | 91.81 | 25.17 | 22.25 | 0.927 | 37.19 | 28.78 | — | — | — | — |
| MambaIR (Guo et al., 2024) | ECCV24 | 22.23 | 0.863 | 63.39 | 20.17 | 21.15 | 0.857 | 56.09 | 24.46 | 25.75 | 0.958 | 19.75 | 20.37 | 21.14 | 0.656 | 154.76 | 32.72 |
| Reti-Diff | Ours | 25.35 | 0.866 | 49.14 | 17.75 | 22.97 | 0.858 | 43.18 | 23.66 | 27.53 | 0.951 | 13.26 | 15.77 | 25.53 | 0.692 | 51.66 | 25.58 |

Table 1: Results on the LLIE task. The best two results are in **red** and **blue** fonts, respectively.

where $\boldsymbol{\mu}^t(\mathbf{Z}_{\mathbf{R}}^t, \mathbf{Z}_{\mathbf{R}}^0) = \frac{1}{\sqrt{\alpha^t}}(\mathbf{Z}_{\mathbf{R}}^t - \frac{1-\alpha^t}{\sqrt{1-\bar{\alpha}^t}}\boldsymbol{\epsilon})$ and $(\boldsymbol{\sigma}^t)^2 = \frac{1-\bar{\alpha}^{t-1}}{1-\bar{\alpha}^t}\beta^t$. $\boldsymbol{\epsilon}$ is the noise in $\mathbf{Z}_{\mathbf{R}}^t$ and we employ a denoising network $\boldsymbol{\epsilon}_\theta(\cdot)$ to estimate $\theta$. To operate in the latent space, we further introduce another RPE module $\widetilde{\text{RPE}}(\cdot)$ to extract the conditional reflectance vector $\mathbf{V_R} \in \mathbb{R}^{3C'}$ from the reflectance image $\mathbf{R}_{LQ}$ of the LQ image, *i.e.*, $\mathbf{V_R} = \widetilde{\text{RPE}}(\text{down}(\mathbf{R}_{LQ}))$. Therefore, the denoising network can be represented by $\boldsymbol{\epsilon}_\theta(\mathbf{Z}_{\mathbf{R}}^t, \mathbf{V_R}, t)$. By setting the variance to $1 - \alpha^t$, we get

$$\mathbf{Z}_{\mathbf{R}}^{t-1} = \frac{1}{\sqrt{\alpha^t}}(\mathbf{Z}_{\mathbf{R}}^t - \frac{1-\alpha^t}{\sqrt{1-\bar{\alpha}^t}}\boldsymbol{\epsilon}_\theta(\mathbf{Z}_{\mathbf{R}}^t, \mathbf{V_R}, t)) + \sqrt{1-\alpha^t}\boldsymbol{\epsilon}^t, \tag{12}$$

where $\boldsymbol{\epsilon}^t \sim \mathcal{N}(0, \mathbf{I})$. By using Eq. (12) for $T$ iterations, we can get the predicted prior $\hat{\mathbf{Z}}_{\mathbf{R}}$ and use it to guide RGformer for image restoration. Because the size of the predicted prior $\hat{\mathbf{Z}}_{\mathbf{R}} \in \mathbb{R}^{3C'}$ is much smaller than the original reflectance image $\mathbf{R}_{LQ} \in \mathbb{R}^{H \times W \times C}$, RLDM needs much less iterations than those image-level diffusion models (Yi et al., 2023). Thus, we can run the complete $T$ iterations for the prior generation rather than randomly selecting one time step.

**Optimization.** We propose the diffusion loss to restrict the predicted priors $\hat{\mathbf{Z}}_{\mathbf{R}}$ and $\hat{\mathbf{Z}}_{\mathbf{L}}$, generated by two RLDMs with specific weights, to be consistent with those extracted from the ground truth:

$$L_{Dif} = \|\mathbf{Z_R} - \hat{\mathbf{Z}}_{\mathbf{R}}\|_1 + \|\mathbf{Z_L} - \hat{\mathbf{Z}}_{\mathbf{L}}\|_1. \tag{13}$$

For restoration quality, we propose joint training RPE, RGformer, and RLDM with the Phase II loss:

$$L_{P2} = L_{Dif} + \lambda_2 L_{Rec} + \lambda_3 L_R, \tag{14}$$

where $\lambda_2$ and $\lambda_3$ are two hyper-parameters and are set as 1 in this paper. The constraints imposed by $L_R$ and $L_{Dif}$, combined with our approach to extracting Retinex priors in a compact space, ensure the generation of high-quality priors that significantly reduce interference from degraded inputs.

### 3.3 INFERENCE

In the inference phase, given the LQ input $\mathbf{I}_{LQ}$, Reti-Diff first uses $\widetilde{\text{RPE}}$ to extract the conditional vectors $\mathbf{V_R}$ and $\mathbf{V_L}$, and then generates predicted Retinex priors $\hat{\mathbf{Z}}_{\mathbf{R}}$ and $\hat{\mathbf{Z}}_{\mathbf{L}}$ with two RLDMs. Under the guidance of the Retinex priors, RGformer generates the restored HQ image $\mathbf{I}_{HQ}$. Benefiting from our Retinex-based diffusion framework, $\mathbf{I}_{HQ}$ enjoys richer texture details and more harmonized illumination, presenting visual-appealing results and further enhancing downstream tasks.

## 4 EXPERIMENT

### 4.1 EXPERIMENTAL SETUP

Our Reti-Diff is implemented in PyTorch on four RTX4090 GPUs and is optimized by Adam with momentum terms $(0.9, 0.999)$. In phases I and II, we train the network for 300K iterations and the learning rate is initialized as $2 \times 10^{-4}$ and gradually reduced to $1 \times 10^{-6}$ with the cosine annealing (Loshchilov, 2016). Random rotation and flips are used for augmentation. Reti-Diff comprises RLDM and RGformer. For RLDM, the channel number $C'$ and the total time step $T$ are set as 64 and 4. $\beta^{1:T}$ linearly increase from $\beta^1 = 0.1$ to $\beta^T = 0.99$. RGformer adopts a 4-level cascade structure. We set the number of transformer blocks, the attention heads, the channel number as $[3, 3, 3, 3]$, $[1, 2, 4, 8]$, $[64, 128, 256, 512]$ from level 1 to 4. We abandon GT-mean for fairness.

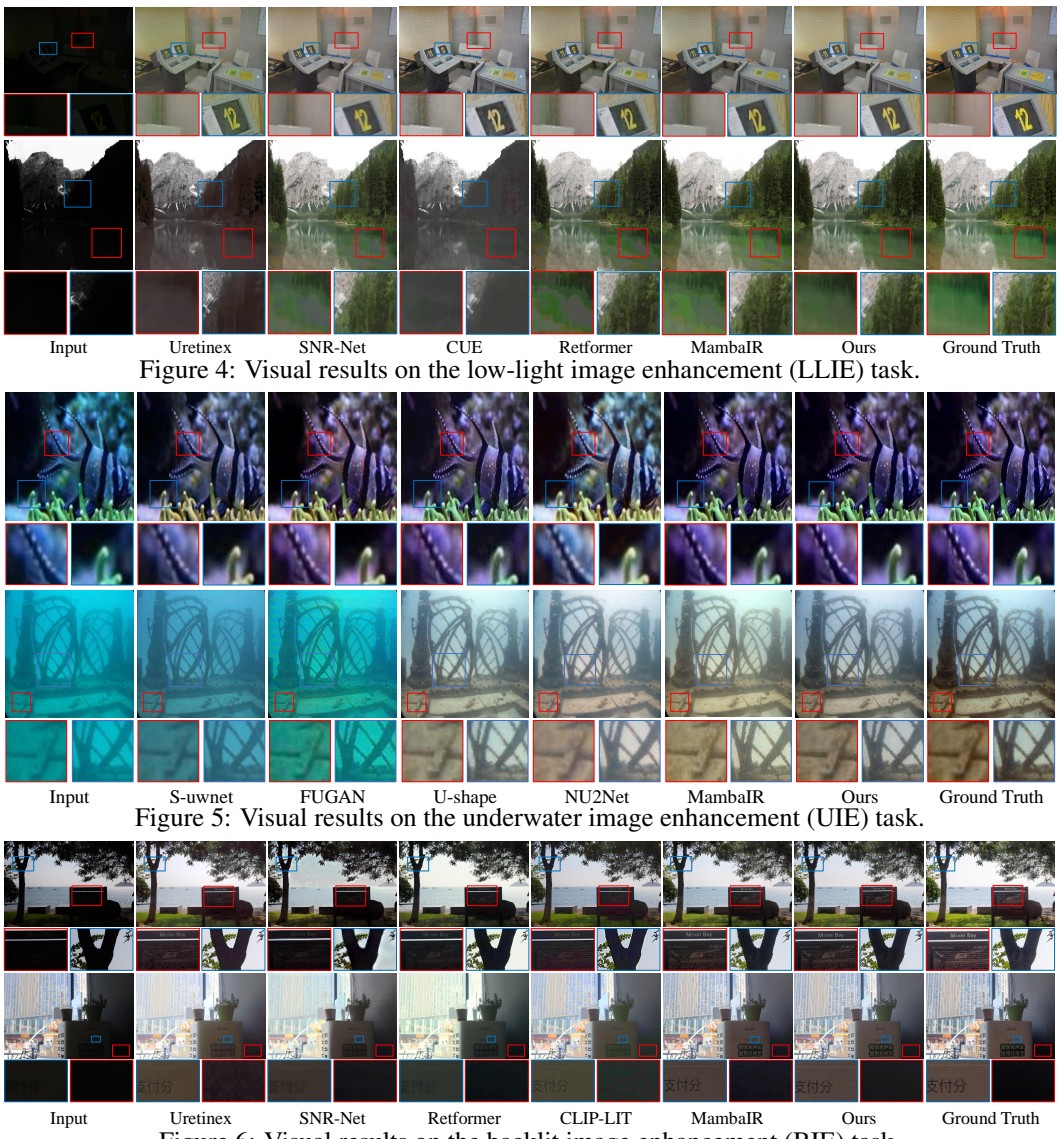

Figure 4: Visual results on the low-light image enhancement (LLIE) task.

Figure 5: Visual results on the underwater image enhancement (UIE) task.

Figure 6: Visual results on the backlit image enhancement (BIE) task.

## 4.2 COMPARATIVE EVALUATION

**Low-light Image Enhancement.** We conduct experiments on four datasets: *LOL-v1* (Wei et al., 2018), *LOL-v2-real* (Yang et al., 2021), *LOL-v2-syn* (Yang et al., 2021), and *SID* (Chen et al., 2019), and involves four metrics: PSNR, SSIM, FID (Heusel et al., 2017), and BIQE (Moorthy & Bovik, 2010). Larger PSNR and SSIM, as well as smaller FID and BIQE, denote superior results. Adhering to the training manner in (Cai et al., 2023), we compare our method against 17 cutting-edge techniques and report the results in Table 1. As depicted in Table 1, our method emerges as the top performer across all datasets, surpassing the second-best method (Diff-Retinex) by 13.2%, underscoring our superiority. Fig. 4 presents qualitative results, showcasing our capacity to generate restored images with corrected illumination and enhanced texture, even in extremely challenging conditions. In contrast, existing methods struggle to address these challenges, such as the boundaries of power lines, color distribution of lakes, and textures of wooded areas. Besides, we also compare the efficiency of the diffusion model-based methods with the size of $256 \times 256$. As shown in Table 2, our Reti-

| Metrics | Diff-Retinex | PyDiff | GSAD | Ours |
|---|---|---|---|---|
| Parameter (M) | 56.88 | 97.89 | **17.17** | 26.11 |
| MACs (G) | 396.32 | 459.69 | 1340.63 | **156.55** |
| FPS | 4.25 | 3.63 | 2.33 | **12.27** |

Table 2: Efficiency analysis in diffusion model-based methods.

Diff has the lowest MACs, highest FPS, and the second smallest parameters. This efficiency can be attributed to our utilization of the diffusion model within a low-dimensional compact latent space. For fairness, results from the compared methods are generated by their provided models.

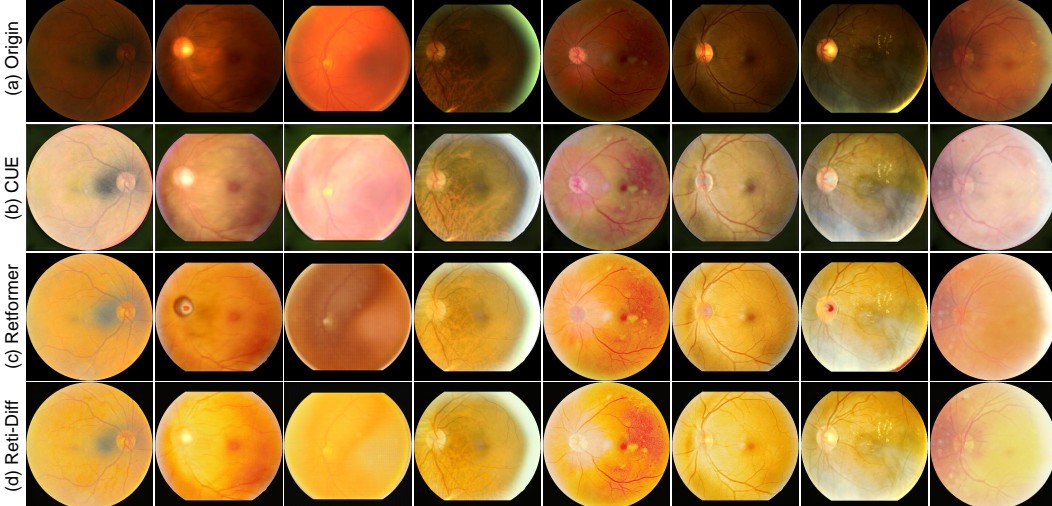

Figure 7: Visual results on real-world fundus images, where we employ the pre-trained model from the *LOL-v1* dataset for inference. Our Reti-Diff presents enhanced details with less color distortion.

| Methods | Sources | UIEB | | | | LSUI | | | |
|---|---|---|---|---|---|---|---|---|---|
| | | PSNR ↑ | SSIM ↑ | UCIQE ↑ | UIQM ↑ | PSNR ↑ | SSIM ↑ | UCIQE ↑ | UIQM ↑ |
| FUGAN (Islam et al., 2020) | IRAL20 | 17.41 | 0.842 | 0.527 | 2.614 | 22.16 | 0.837 | 0.576 | 2.667 |
| EnGAN (Jiang et al., 2021) | TIP21 | 17.73 | 0.833 | 0.529 | 2.465 | 19.30 | 0.851 | 0.587 | 2.817 |
| Ucolor (Li et al., 2021) | TIP21 | 20.78 | 0.868 | 0.537 | 3.049 | 22.91 | 0.886 | 0.594 | 2.735 |
| S-uwnet (Naik et al., 2021) | AAAI21 | 18.28 | 0.855 | 0.544 | 2.942 | 20.89 | 0.875 | 0.582 | 2.746 |
| PUIE (Fu et al., 2022) | ECCV22 | 21.38 | 0.882 | 0.566 | 3.021 | 23.70 | 0.902 | 0.605 | 2.974 |
| U-shape (Peng et al., 2023) | TIP23 | 22.91 | 0.905 | 0.592 | 2.896 | 24.16 | 0.917 | 0.603 | 3.022 |
| PUGAN (Cong et al., 2023) | TIP23 | 23.05 | 0.897 | 0.608 | 2.902 | 25.06 | 0.916 | 0.629 | 3.106 |
| ADP (Zhou et al., 2023b) | IJCV23 | 22.90 | 0.892 | 0.621 | 3.005 | 24.28 | 0.913 | 0.626 | 3.075 |
| NU2Net (Guo et al., 2023) | AAAI23 | 22.38 | 0.903 | 0.587 | 2.936 | 25.07 | 0.908 | 0.615 | 3.112 |
| AST (Zhou et al., 2024) | CVPR24 | 22.19 | 0.908 | 0.602 | 2.981 | 27.46 | 0.916 | 0.632 | 3.107 |
| MambaIR (Guo et al., 2024) | ECCV24 | 22.60 | 0.939 | 0.617 | 2.991 | 27.68 | 0.916 | 0.630 | 3.118 |
| Reti-Diff | Ours | 24.12 | 0.910 | 0.631 | 3.088 | 28.10 | 0.929 | 0.646 | 3.208 |

Table 3: Results on the UIE task.

| Methods | Sources | BAID | | | |
|---|---|---|---|---|---|
| | | PSNR ↑ | SSIM ↑ | LPIPS ↓ | FID ↓ |
| EnGAN (Jiang et al., 2021) | TIP21 | 17.96 | 0.819 | 0.182 | 43.55 |
| RUAS (Liu et al., 2021) | CVPR21 | 18.92 | 0.813 | 0.262 | 40.07 |
| URetinex (Wu et al., 2022) | CVPR22 | 19.08 | 0.845 | 0.206 | 42.26 |
| SNR-Net (Xu et al., 2022) | CVPR22 | 20.86 | 0.860 | 0.213 | 39.73 |
| Restormer (Zamir et al., 2022) | CVPR22 | 21.07 | 0.832 | 0.192 | 41.17 |
| Retformer (Cai et al., 2023) | ICCV23 | 22.03 | 0.862 | 0.173 | 45.27 |
| CLIP-LIT (Liang et al., 2023) | ICCV23 | 21.13 | 0.853 | 0.159 | 37.30 |
| Diff-Retinex (Yi et al., 2023) | ICCV23 | 22.07 | 0.861 | 0.160 | 38.07 |
| DiffIR (Xia et al., 2023) | ICCV23 | 21.10 | 0.835 | 0.175 | 40.35 |
| AST (Zhou et al., 2024) | CVPR24 | 22.61 | 0.851 | 0.156 | 32.47 |
| MambaIR (Guo et al., 2024) | ECCV24 | 23.07 | 0.874 | 0.153 | 29.13 |
| Reti-Diff | Ours | 23.19 | 0.876 | 0.147 | 27.47 |

Table 4: Results on the BIE task.

| Methods | Sources | DICM | | LIME | | MEF | | NPE | | VV | |
|---|---|---|---|---|---|---|---|---|---|---|---|
| | | PI ↓ | NIQE ↓ | PI ↓ | NIQE ↓ | PI ↓ | NIQE ↓ | PI ↓ | NIQE ↓ | PI ↓ | NIQE ↓ |
| EnGAN (Jiang et al., 2021) | TIP21 | 4.173 | 4.064 | 3.669 | 4.593 | 4.015 | 4.705 | 3.226 | 3.993 | 3.386 | 4.047 |
| KinD++ (Zhang et al., 2021b) | IJCV21 | 3.835 | 3.898 | 3.785 | 4.908 | 4.016 | 4.557 | 3.179 | 3.915 | 3.773 | 3.822 |
| SNR-Net (Xu et al., 2022) | CVPR22 | 3.585 | 4.715 | 3.753 | 5.937 | 3.677 | 6.449 | 3.278 | 6.446 | 3.503 | 9.506 |
| DCC-Net (Zhang et al., 2022) | CVPR22 | 3.630 | 3.709 | 3.312 | 4.425 | 3.424 | 4.598 | 2.878 | 3.706 | 3.615 | 3.286 |
| UHDFor (Li et al., 2023) | ICLR23 | 3.684 | 4.575 | 4.124 | 4.430 | 3.813 | 4.231 | 3.135 | 3.867 | 3.319 | 4.330 |
| PairLIE (Fu et al., 2023) | CVPR23 | 3.685 | 4.034 | 3.387 | 4.587 | 4.133 | 4.065 | 3.726 | 4.187 | 3.334 | 3.574 |
| GDP (Fei et al., 2023) | CVPR23 | 3.552 | 4.358 | 4.115 | 4.891 | 3.694 | 4.609 | 3.097 | 4.032 | 3.431 | 4.683 |
| GSAD (Jinhui et al., 2023) | NIPS23 | — | 3.465 | — | 4.517 | — | 3.815 | — | 3.806 | — | 3.355 |
| Reti-Diff | Ours | 2.351 | 3.255 | 2.837 | 3.693 | 3.308 | 3.792 | 2.599 | 3.384 | 3.341 | 3.000 |

Table 5: Results on the real-world IDIR task.

| Datasets | | L-v2-r | | L-v2-s | |
|---|---|---|---|---|---|
| Metrics | | PSNR ↑ | SSIM ↑ | PSNR ↑ | SSIM ↑ |
| RLDM | w/o RLDM | 21.25 | 0.822 | 25.38 | 0.918 |
| | w/o DM | 21.72 | 0.830 | 25.83 | 0.927 |
| RGformer | w/o DFA | 22.26 | 0.840 | 26.49 | 0.925 |
| | w/o RG-MCA | 21.73 | 0.840 | 25.92 | 0.913 |
| | w/o $D_a(\cdot)$ | 22.58 | 0.847 | 26.80 | 0.944 |
| Train | w/o joint | 22.83 | 0.853 | 27.18 | 0.947 |
| Reti-Diff (Ours) | | 22.97 | 0.858 | 27.53 | 0.951 |

Table 6: Break down ablation.

**Underwater Image Enhancement.** We select two widely-used underwater image enhancement datasets: *UIEB* (Li et al., 2019) and *LSUI* (Peng et al., 2023). Following (Guo et al., 2023), we employ two metrics tailored for underwater images, namely UCIQE (Yang & Sowmya, 2015) and UIQM (Panetta et al., 2015). In all cases, higher values indicate better performance. The results are presented in Table 3. As shown in Table 3, our method achieves the highest performance and outperforms the second-best method (MambaIR) by 2.30%. A qualitative analysis is presented in Fig. 5, illustrating our capacity to correct underwater color aberrations and highlight texture details.

**Backlit Image Enhancement.** Following CLIP-LIT (Liang et al., 2023), we select the *BAID* (Lv et al., 2022) dataset for network training. Apart from PSNR and SSIM, our evaluation also selects two perception metrics: LPIPS (Zhang et al., 2018) and FID (Heusel et al., 2017), where lower values denote better performance. We report our results in Table 4. As demonstrated in Table 4, our method outperforms all other methods across all metrics. Besides, a visual comparison in Fig. 6 provides additional evidence of our superiority in detail reconstruction and color correction.

**Real-world Illumination Degradation Image Restoration.** We also explore our applicability in real-world IDIR tasks. Following CIDNet (Feng et al., 2024), we selected five commonly-used real-world datasets, *i.e.*, *DICM* (Lee et al., 2013), *LIME* (Guo et al., 2016), *MEF* (Wang et al., 2013), *NPE* (Ma et al., 2015), and *VV* (He et al., 2024b), with only low-quality images available. Therefore, akin to (Feng et al., 2024), we leverage the model pretrained on *LOL-v2-syn* for inference and select PI (Blau et al., 2018) and NIQE (Mittal et al., 2012) as evaluation metrics, where lower scores indicate better results. As presented in Table 5, our method achieves optimal results and surpasses the second-based method (DCC-Net (Zhang et al., 2022)) by 13.39%.

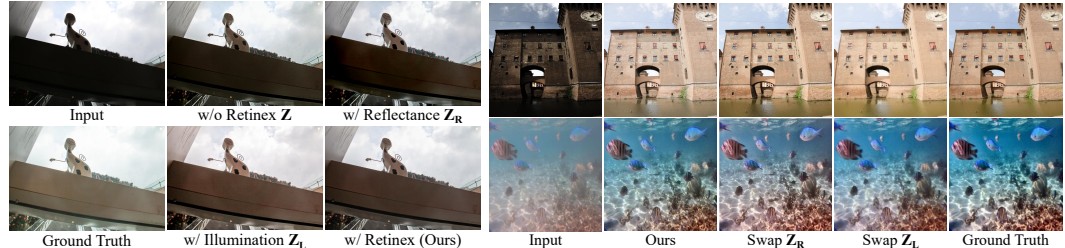

(a) Break down ablation in Retinex priors.  (b) Swap our Retinex priors with that extracted from GT.

Figure 8: Visual validation of the effectiveness of Retinex priors.

| Dataset | Metrics | All data | | | | | | Extreme condition (similar in $\mathbf{Z_R}$) | | | | Extreme condition (similar in $\mathbf{Z_L}$) | | | |
| --- | --- | --- | --- | --- | --- | --- | --- | --- | --- | --- | --- | --- | --- | --- | --- |
| | | w/o $\mathbf{Z}$ | w/ $\mathbf{Z_R}$ | w/ $\mathbf{Z_L}$ | w/ $\mathbf{Z}$ (Ours) | Swap $\mathbf{Z_R}$ | Swap $\mathbf{Z_L}$ | w/o $\mathbf{Z}$ | w/ $\mathbf{Z_R}$ | w/ $\mathbf{Z_L}$ | w/ $\mathbf{Z}$ (Ours) | w/o $\mathbf{Z}$ | w/ $\mathbf{Z_R}$ | w/ $\mathbf{Z_L}$ | w/ $\mathbf{Z}$ (Ours) |
| *L-v2-r* | PSNR | 21.63 | 22.13 | 22.35 | 22.97 | 23.31 | 23.17 | 21.37 | 21.66 | 22.57 | 23.57 | 21.19 | 23.06 | 21.85 | 24.06 |
| | SSIM | 0.830 | 0.842 | 0.839 | 0.858 | 0.862 | 0.863 | 0.823 | 0.834 | 0.843 | 0.862 | 0.820 | 0.845 | 0.831 | 0.868 |
| *L-v2-s* | PSNR | 26.25 | 26.62 | 27.02 | 27.53 | 27.92 | 27.75 | 25.68 | 26.17 | 27.05 | 28.57 | 25.42 | 27.34 | 26.24 | 28.80 |
| | SSIM | 0.939 | 0.945 | 0.941 | 0.951 | 0.957 | 0.956 | 0.922 | 0.930 | 0.951 | 0.966 | 0.920 | 0.958 | 0.936 | 0.965 |

Table 7: Effect of Retinex priors in all data and two extreme conditions (each with 10 images).

| Datasets | Metrics | Ufo | Ufo+RLDM | Res | Res+RLDM | Ret | Ret+RLDM |
| --- | --- | --- | --- | --- | --- | --- | --- |
| *L-v2-r* | PSNR | 18.82 | 21.37 | 19.94 | 21.56 | 22.80 | 23.16 |
| | SSIM | 0.771 | 0.794 | 0.827 | 0.837 | 0.840 | 0.849 |
| | Gain | – | 8.27% | – | 4.67% | – | 1.33% |
| *L-v2-s* | PSNR | 19.66 | 22.08 | 21.41 | 24.15 | 25.67 | 26.81 |
| | SSIM | 0.871 | 0.889 | 0.830 | 0.862 | 0.930 | 0.942 |
| | Gain | – | 7.19% | – | 8.33% | – | 2.87% |

Table 8: Generalization of Retinex priors. "Ufo", "Res", and "Ret" are Uformer, Restormer, and Retformer.

Figure 9: Ablation study of the number of iterations in RLDM on *LOL-v2-syn*.

## 4.3 ABLATION STUDY

**Effect of RLDM.** As shown in Tables 6 and 7, we ablate RLDM by directly removing RLDM, replacing the diffusion model with a linear model that shares the same structure with the denoising network (w/o DM), or retraining RLDM in the RGB domain, *i.e.*, w/o $\mathbf{Z}$, rather than in the reflectance and illumination domain (RGformer is guided by one RGB prior instead). The three changes bring significant performance drops, underscoring the critical role of RLDM in enhancing the restoration process and the importance of using the diffusion model to extract compact priors.

**Effect of RGformer.** We analyze the impact of our RGformer by removing key modules, such as DFA, RG-MCA, and the auxiliary decoder $D_a(\cdot)$. As shown in Table 6, the outcomes indicate performance decreases when these modules are removed, highlighting their essential roles. Additionally, we also conduct an evaluation to affirm the significance of joint training in our method.

**Effect of Retinex priors.** We explore the effect of our Retinex prior from three aspects: **(1)** We conduct the break down ablation for the Retinex priors and report the results in Fig. 8 and Table 7. These findings demonstrate the effect of our Reflectance prior $\mathbf{Z_R}$ in detail enhancement and our Illumination prior $\mathbf{Z_L}$ in illumination correction. **(2)** We then swap our Retinex priors with those extracted from ground truth. As shown in Fig. 8 and Table 7, the results guided by the swapped ground-truth priors exhibit limited performance gains. This indicates our RLDM can already generate high-quality priors, which is attributed to the constraints in Eqs. 7 and 13 and our approach to extracting Retinex priors in a compact space, significantly reducing interference from degraded inputs. **(3)** We further explore the potential of Retinex priors under extreme conditions where the reflectance or illumination priors exhibit high similarity between low-quality and ground-truth images. To validate this, five human subjects rated the similarity of the Retinex priors between low-quality and ground-truth images. Two sets of images, each with 10 images, were selected based on the highest similarity in reflectance and illumination priors. The results in Table 7 verify the effect of our Retinex priors even in this condition. This is attributed to the generative capacity of our RLDM and the information aggregation capacity of our RGformer.

**Generalization of Retinex priors.** To assess our generalizability, we incorporate our RLDM into existing cutting-edge methods, namely Ufo (Uformer (Wang et al., 2022)), Res (Restormer (Zamir et al., 2022)) and Ret (Retformer (Cai et al., 2023)), and use the extracted Retinex priors to guide these methods for image enhancement, where the training settings are kept consistent with Reti-Diff. The results are shown in Table 8. Table 8 reveals that RLDM significantly improves the performance of all frameworks, indicating the strong generalization capabilities of our Retinex priors.

| Methods | L-v1 | L-v2-r | L-v2-s | SID | Mean |
|---|---|---|---|---|---|
| KinD | 2.31 | 2.25 | 2.46 | 2.33 | 2.34 |
| EnGAN | 2.63 | 1.69 | 2.23 | 1.24 | 1.95 |
| RUAS | 3.57 | 3.06 | 3.01 | 2.23 | 2.97 |
| Restormer | 3.26 | 3.32 | 3.41 | 2.53 | 3.13 |
| Uretinex | 3.82 | 3.98 | 3.70 | 3.28 | 3.70 |
| SNR-Net | 3.76 | 4.12 | 3.58 | 3.42 | 3.72 |
| CUE | 3.62 | 3.81 | 3.28 | 3.09 | 3.45 |
| Retformer | 3.35 | 4.02 | 3.71 | 3.35 | 3.61 |
| Ours | 4.05 | 4.33 | 3.92 | 3.75 | 4.01 |

Table 9: User study.

| Methods (AP) | Bicycle | Boat | Bottle | Bus | Car | Cat | Chair | Cup | Dog | Motor | People | Table | Mean |
|---|---|---|---|---|---|---|---|---|---|---|---|---|---|
| Baseline | 74.7 | 64.9 | 70.7 | 84.2 | 79.7 | 47.3 | 58.6 | 67.1 | 64.1 | 66.2 | 73.9 | 45.7 | 66.4 |
| RetinexNet | 72.8 | 66.4 | 67.3 | 87.5 | 80.6 | 52.8 | 60.0 | 67.8 | 68.5 | 69.3 | 71.3 | 46.2 | 67.5 |
| KinD | 73.2 | 67.1 | 64.6 | 86.8 | 79.5 | 58.7 | 63.4 | 67.5 | 67.4 | 62.3 | 75.5 | 51.4 | 68.1 |
| MIRNet | 74.9 | 69.7 | 68.3 | 89.7 | 77.6 | 57.8 | 56.9 | 66.4 | 69.7 | 64.6 | 74.6 | 53.4 | 68.6 |
| RUAS | 75.7 | 71.2 | 73.5 | 90.7 | 80.1 | 59.3 | 67.0 | 66.3 | 68.3 | 66.9 | 72.6 | 50.6 | 70.2 |
| Restormer | 77.0 | 71.0 | 68.8 | 91.6 | 77.1 | 62.5 | 57.3 | 68.0 | 69.6 | 69.2 | 74.6 | 49.7 | 69.7 |
| SCI | 73.4 | 68.0 | 69.5 | 86.2 | 74.5 | 63.1 | 59.5 | 61.0 | 67.3 | 63.9 | 73.2 | 47.3 | 67.2 |
| SNR-Net | 78.3 | 74.2 | 74.5 | 89.6 | 82.7 | 66.8 | 66.3 | 62.5 | 74.7 | 63.1 | 73.3 | 57.2 | 71.9 |
| Retformer | 78.1 | 74.5 | 74.2 | 91.2 | 82.2 | 65.0 | 63.3 | 67.0 | 75.4 | 68.6 | 75.3 | 55.6 | 72.5 |
| Ours | 82.0 | 77.9 | 76.4 | 92.2 | 83.3 | 69.6 | 67.4 | 74.4 | 75.5 | 74.3 | 78.3 | 57.9 | 75.8 |

Table 10: Low-light image detection on *ExDark*.

| Methods (IoU) | Bicycle | Boat | Bottle | Bus | Car | Cat | Chair | Dog | Horse | People | Mean |
|---|---|---|---|---|---|---|---|---|---|---|---|
| Baseline | 43.5 | 36.3 | 48.6 | 70.5 | 67.3 | 46.6 | 11.2 | 42.4 | 56.7 | 57.8 | 48.1 |
| RetinexNet | 48.6 | 41.7 | 51.7 | 77.6 | 68.3 | 52.7 | 15.8 | 46.3 | 60.2 | 62.3 | 52.5 |
| KinD | 51.3 | 40.2 | 53.2 | 76.8 | 69.4 | 50.8 | 14.6 | 47.3 | 60.3 | 60.9 | 52.5 |
| MIRNet | 50.3 | 42.9 | 47.4 | 73.6 | 62.7 | 50.4 | 15.8 | 46.3 | 61.0 | 63.3 | 51.4 |
| RUAS | 53.0 | 37.3 | 50.4 | 71.3 | 72.3 | 47.6 | 15.9 | 50.8 | 63.6 | 60.8 | 52.3 |
| Restormer | 53.8 | 43.8 | 51.4 | 68.7 | 66.8 | 52.6 | 21.6 | 54.8 | 59.8 | 63.3 | 53.7 |
| SCI | 54.5 | 46.3 | 57.2 | 78.4 | 73.3 | 49.1 | 22.8 | 49.0 | 62.1 | 66.9 | 56.0 |
| SNR-Net | 57.7 | 48.6 | 59.5 | 81.3 | 74.8 | 50.2 | 24.4 | 50.7 | 64.3 | 68.7 | 58.0 |
| Retformer | 50.9 | 47.7 | 58.6 | 77.2 | 68.1 | 53.2 | 17.4 | 52.0 | 61.3 | 71.5 | 55.8 |
| Ours | 59.8 | 51.5 | 62.1 | 85.5 | 76.6 | 57.7 | 28.9 | 56.3 | 66.2 | 73.4 | 61.8 |

| Methods | COD10K | | | | NC4K | | | |
|---|---|---|---|---|---|---|---|---|
| | $M \downarrow$ | $F_\beta \uparrow$ | $E_\phi \uparrow$ | $S_\alpha \uparrow$ | $M \downarrow$ | $F_\beta \uparrow$ | $E_\phi \uparrow$ | $S_\alpha \uparrow$ |
| Baseline | 0.050 | 0.625 | 0.812 | 0.756 | 0.071 | 0.733 | 0.816 | 0.763 |
| RetinexNet | 0.041 | 0.667 | 0.845 | 0.789 | 0.055 | 0.750 | 0.842 | 0.819 |
| KinD | 0.039 | 0.673 | 0.849 | 0.792 | 0.052 | 0.762 | 0.875 | 0.822 |
| MIRNet | 0.037 | 0.697 | 0.857 | 0.799 | 0.049 | 0.802 | 0.888 | 0.833 |
| RUAS | 0.036 | 0.705 | 0.861 | 0.803 | 0.051 | 0.795 | 0.883 | 0.827 |
| Restormer | 0.036 | 0.700 | 0.859 | 0.800 | 0.050 | 0.792 | 0.880 | 0.830 |
| SCI | 0.037 | 0.710 | 0.863 | 0.805 | 0.051 | 0.782 | 0.880 | 0.836 |
| SNR-Net | 0.036 | 0.703 | 0.865 | 0.803 | 0.049 | 0.801 | 0.892 | 0.838 |
| Retformer | 0.037 | 0.682 | 0.861 | 0.806 | 0.052 | 0.766 | 0.881 | 0.832 |
| Ours | 0.034 | 0.725 | 0.880 | 0.813 | 0.047 | 0.804 | 0.897 | 0.841 |

Table 11: Low-light semantic segmentation, where images are darkened by (Zhang et al., 2021a).

Table 12: Low-light concealed object segmentation.

## 4.4 USER STUDY AND DOWNSTREAM TASKS

**User Study.** We conduct a user study to assess the subjective visual perception of low-light image enhancement. In this study, 29 human subjects are invited to assign scores to the enhanced results based on four criteria: (1) The presence of underexposed or overexposed regions. (2) The existence of color distortion. (3) The occurrence of undesired noise or artifacts. (4) The inclusion of essential structural details. Participants rate the results on a scale from 1 (worst) to 5 (best). Each low-light image is presented alongside its enhanced results, with the names of the enhancement methods concealed. The scores are reported in Table 9, where our method receives the highest scores across all four datasets. This highlights our effectiveness in generating visually appealing results.

**Low-light Object Detection.** The enhanced images are expected to have better downstream performance. We first verify this on low-light object detection. Following (Cai et al., 2023), all compared methods are performed on *ExDark* (Loh & Chan, 2019) with YOLO, which is retrained from scratch with their own enhanced results. The "Baseline" represents the performance on low-quality images without enhancement. As shown in Table 10, our Reti-Diff exhibits a substantial advantage over existing methods and our performance surpasses that of the second-best method, Retformer, by 4.72%, verifying our efficacy in facilitating high-level vision understanding.

**Low-light Image Segmentation.** We also conducted segmentation tasks and retrained the segmentor for each method following that in detection. **(1)** For semantic segmentation, following (Ju et al., 2022), we apply image darkening to samples from the *VOC* (Everingham et al., 2010) dataset according to (Zhang et al., 2021a). We then employ Mask2Former (Cheng et al., 2022) to segment the enhanced results of these darkened images and select Intersection over Union (IoU) for evaluation. As shown in Table 11, we achieve the highest performance across all classes, surpassing the second-best method by 6.55%. **(2)** We further venture into concealed object segmentation (COS) on two datasets, *COD10K* (Fan et al., 2021) and *NC4K* (Lv et al., 2021), which is a challenging task aimed at delineating objects with inherent background similarity. We also apply image darkening and enlist FEDER (He et al., 2023c) to segment the enhanced results. We evaluate the results using four metrics: mean absolute error $(M)$, adaptive F-measure $(F_\beta)$, mean E-measure $(E_\phi)$, and structure measure $(S_\alpha)$. As depicted in Table 12, our method exhibits superior performance compared to the second-best method, SNR-Net, with a margin of 2.16% on average.

## 5 CONCLUSIONS

To balance generation capability and computational efficiency, our approach adopts DM within a compact latent space to generate guidance priors. Specifically, we introduce RLDM to extract Retinex priors, which are subsequently supplied to RGformer for feature decomposition, ensuring precise detailed reconstruction and effective illumination correction. RGformer then refines and aggregates the decomposed features, enhancing the robustness in handling complex degradation scenes. Our approach is validated through extensive experiments, establishing clear superiority.

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

# Supplementary Materials for
# Reti-Diff: Illumination Degradation Image Restoration with Retinex-based Latent Diffusion Model

## Contents

## A  Methodology

### A.1  Retinex-based Latent Diffusion Model

In this section, we provide a detailed derivation for $\hat{\mathbf{Z}}_{\mathbf{L}}$.

**Diffusion process.** In the diffusion process, we first use the pretrained RPE to extract the reflectance prior $\mathbf{Z}_{\mathbf{L}}$, which is treated as the starting point of the forward Markov process, *i.e.*, $\mathbf{Z}_{\mathbf{L}} = \mathbf{Z}_{\mathbf{L}}^0$. We then gradually add Gaussian noise to $\mathbf{Z}_{\mathbf{L}}$ by $T$ iterations and each iteration can be defined as:

$$q\left(\mathbf{Z}_{\mathbf{L}}^t | \mathbf{Z}_{\mathbf{L}}^{t-1}\right) = \mathcal{N}\left(\mathbf{Z}_{\mathbf{L}}^t; \sqrt{1 - \beta^t}\mathbf{Z}_{\mathbf{L}}^{t-1}, \beta^t\mathbf{I}\right), \tag{1}$$

where $t = 1, \cdots, T$. $\mathbf{Z}_{\mathbf{L}}^t$ denotes the noisy prior at time step $t$, $\beta^t$ is the predefined factor that controls the noise variance, and $\mathcal{N}$ is the Gaussian distribution. Following (Kingma & Welling, 2013), Eq. (1) can be simplified as follows:

$$q\left(\mathbf{Z}_{\mathbf{L}}^t | \mathbf{Z}_{\mathbf{L}}^0\right) = \mathcal{N}\left(\mathbf{Z}_{\mathbf{L}}^t; \sqrt{\bar{\alpha}^t}\mathbf{Z}_{\mathbf{L}}^0, (1 - \bar{\alpha}^t)\mathbf{I}\right), \tag{2}$$

where $\alpha^t = 1 - \beta^t$ and $\bar{\alpha}^t = \prod_{i=1}^t \alpha^i$.

**Reverse process.** In the reverse process, RLDM aims to extract the reflectance prior from pure Gaussian noise. Thus, RLDM samples a Gaussian random noise map $\mathbf{Z}_{\mathbf{L}}^T$ and then gradually denoise it to run backward from $\mathbf{Z}_{\mathbf{L}}^T$ to $\mathbf{Z}_{\mathbf{L}}^0$:

$$p\left(\mathbf{Z}_{\mathbf{L}}^{t-1} | \mathbf{Z}_{\mathbf{L}}^t, \mathbf{Z}_{\mathbf{L}}^0\right) = \mathcal{N}\left(\mathbf{Z}_{\mathbf{L}}^{t-1}; \boldsymbol{\mu}^t(\mathbf{Z}_{\mathbf{L}}^t, \mathbf{Z}_{\mathbf{L}}^0), (\sigma^t)^2\mathbf{I}\right), \tag{3}$$

where mean $\boldsymbol{\mu}^t(\mathbf{Z}_{\mathbf{L}}^t, \mathbf{Z}_{\mathbf{L}}^0) = \frac{1}{\sqrt{\alpha^t}}(\mathbf{Z}_{\mathbf{L}}^t - \frac{1-\alpha^t}{\sqrt{1-\bar{\alpha}^t}}\boldsymbol{\epsilon})$ and variance $(\sigma^t)^2 = \frac{1-\bar{\alpha}^{t-1}}{1-\bar{\alpha}^t}\beta^t$. $\boldsymbol{\epsilon}$ denotes the noise in $\mathbf{Z}_{\mathbf{L}}^t$ and is the only uncertain variable. Following previous practice (Xia et al., 2023), we employ a denoising network $\boldsymbol{\epsilon}_\theta(\cdot)$ to estimate $\theta$. To operate in the latent space, we further introduce another RPE module $\widetilde{\mathrm{RPE}}(\cdot)$ to extract the conditional reflectance vector $\mathbf{V}_{\mathbf{L}} \in \mathbb{R}^{C'}$ from the reflectance image $\mathbf{L}_{LQ}$ of the LQ image, *i.e.*, $\mathbf{V}_{\mathbf{L}} = \widetilde{\mathrm{RPE}}(\mathrm{down}(\mathbf{L}_{LQ}))$. Therefore, the denoising

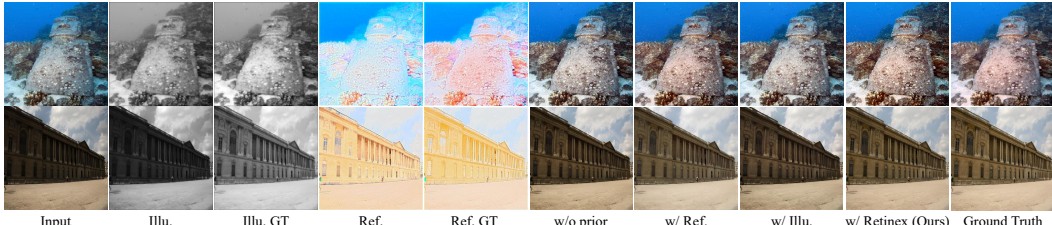

| Input | Illu. | Illu. GT | Ref. | Ref. GT | w/o prior | w/ Ref. | w/ Illu. | w/ Retinex (Ours) | Ground Truth |

Fig. S1: Effect of Retinex priors in extreme conditions, where the two rows share a similarity in reflectance and illumination components, respectively.

| Datasets | Metrics | $\ell_2$-norm | $\ell_1$-norm (Ours) |
|---|---|---|---|
| L-v2-s | PSNR | 27.26 | 27.53 |
| | SSIM | 0.949 | 0.951 |
| L-v2-r | PSNR | 22.62 | 22.97 |
| | SSIM | 0.853 | 0.858 |

Fig. S1: Effect of $\ell_p$-norm in Loss Functions.

| Datasets | Metrics | $\lambda_1 = 0.1$ | $\lambda_1 = 1$ (Ours) | $\lambda_1 = 10$ | $\lambda_2 = 0.1$ | $\lambda_2 = 1$ (Ours) | $\lambda_2 = 10$ | $\lambda_3 = 0.1$ | $\lambda_3 = 1$ (Ours) | $\lambda_3 = 10$ |
|---|---|---|---|---|---|---|---|---|---|---|
| L-v2-s | PSNR | 27.15 | 27.53 | 27.33 | 27.08 | 27.53 | 27.33 | 27.26 | 27.53 | 27.35 |
| | SSIM | 0.949 | 0.951 | 0.948 | 0.946 | 0.951 | 0.947 | 0.952 | 0.951 | 0.946 |
| L-v2-r | PSNR | 22.86 | 22.97 | 22.82 | 22.36 | 22.97 | 22.76 | 22.33 | 22.97 | 22.16 |
| | SSIM | 0.857 | 0.858 | 0.855 | 0.851 | 0.858 | 0.856 | 0.853 | 0.858 | 0.850 |

Fig. S2: Effect of $\ell_p$-norm in Loss Functions.

network can be represented by $\epsilon_\theta(\mathbf{Z}_\mathbf{L}^t, \mathbf{V_L}, t)$. By setting the variance to $1 - \alpha^t$, we get

$$\mathbf{Z}_\mathbf{L}^{t-1} = \frac{1}{\sqrt{\alpha^t}}(\mathbf{Z}_\mathbf{L}^t - \frac{1 - \alpha^t}{\sqrt{1 - \bar{\alpha}^t}}\epsilon_\theta(\mathbf{Z}_\mathbf{L}^t, \mathbf{V_L}, t)) + \sqrt{1 - \alpha^t}\epsilon^t, \quad (4)$$

where $\epsilon^t \sim \mathcal{N}(0, \mathbf{I})$.

# B  EXPERIMENT

## B.1  ABLATION STUDY

Following the practice in the manuscript, we select *LOL-v2-real* and *LOL-v2-syn* to conduct ablation studies, where the two datasets are abbreviated as *L-v2-r* and *L-v2-s*.

**Effect of Retinex priors in extreme conditions.** We investigate the potential of Retinex priors, *i.e.*, **ZR** and **ZL**, under extreme conditions where the reflectance or illumination components exhibit high similarity between low-quality and ground-truth images. As shown in Fig. S1, the extracted priors have a diminished effect when the corresponding component shows the similarity between low-quality and ground-truth images. This is because the corresponding component undergoes minimal degradation.

**Effect of $\ell_p$-norm in Loss Functions.** We explore the effect of $\ell_p$-norm in loss functions. As shown in Table S1, Reti-Diff achieves better performance when using $\ell_1$-norm. Therefore, our loss functions select $\ell_1$-norm.

**Parameter Analysis.** Our Reti-Diff is optimized with multiple losses, which are balanced by three hyperparameters, *i.e.*, $\lambda_1$, $\lambda_2$, and $\lambda_3$. To analyze their impact, we vary one of the parameters and fix others, and report the results in Table S2. Overall, we find that the different coefficients in the tested range only slightly influence the final performance and $\lambda_1$, $\lambda_2$, and $\lambda_3$ obtain better results when they are set to 1. So we set those parameters to 1 each.

## B.2  COMPARATIVE EVALUATION

**Low-light Image Enhancement.** As shown in Fig. S2, we provide more visualization results. Our method can generate enhanced images with corrected illumination and enhanced texture, even in extremely challenging conditions.

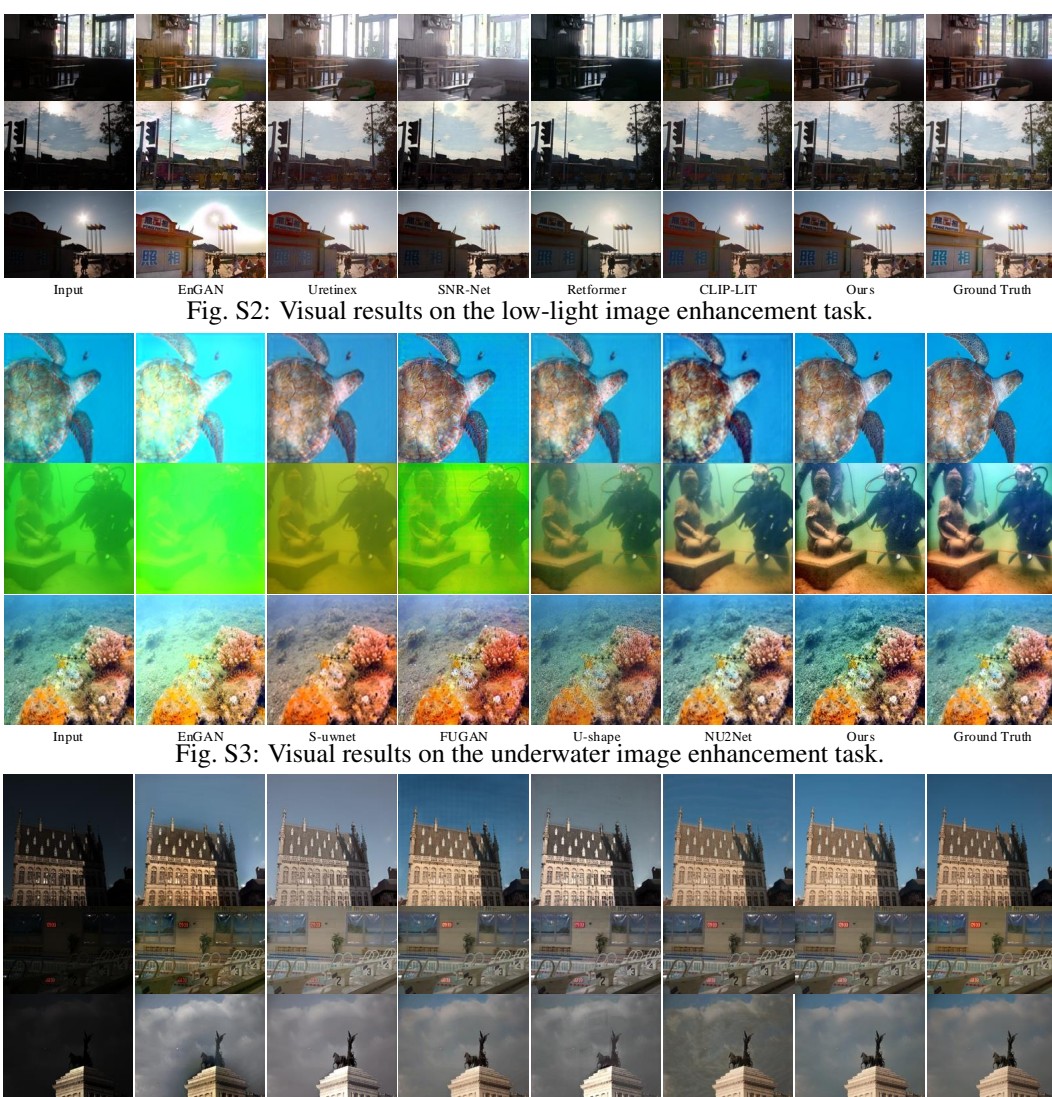

Fig. S2: Visual results on the low-light image enhancement task.

Fig. S3: Visual results on the underwater image enhancement task.

Fig. S4: Visual results on the backlit image enhancement task.

**Underwater Image Enhancement.** More qualitative analyses are presented in Fig. S3, illustrating our superiority in underwater color correction and fine texture details reconstruction.

**Backlit Image Enhancement.** Furthermore, a visual comparison in Fig. S4 provides additional evidence of our superiority in detail reconstruction and color correction. All methods are trained by cropping the training data as $256 \times 256$ for fairness.

## C    DISCUSSIONS

Our Reti-Diff is the first LDM-based solution specifically tailored for the IDIR task, setting it apart from existing LDM-based methods applied in other tasks. To illustrate the distinctions, we compare it with a general enhancement method, DiffIR (Xia et al., 2023): **(1) Motivation.** Reti-Diff targets enhancing details and correcting degraded illumination. Thus, we enable RLDM to learn Retinex knowledge and generate Retinex priors from the low-quality input. We contend that relying solely on priors extracted from the RGB domain struggles to fully represent valuable texture details and correct illumination cues, leading to suboptimal restoration performance. To verify this, we substitute our RLDM for the LDM structure used in DiffIR. In *LOL-v2-syn*, we observe that the PSNR rises from 24.76 to 26.14 and the SSIM increases from 0.921 to 0.933. **(2) Implementation.** Apart from proposing RLDM to extract Retinex priors, we further modify the structure of RGformer to implic-

itly model the Retinex theory at the feature level and introduce an auxiliary decoder to reconstruct the decomposed Retinex components to the RGB domain.

## D  LIMITATIONS AND FUTURE WORK

Our Reti-Diff faces challenges in simultaneously recovering illumination and restoring texture details when the low-quality inputs suffer from severe illumination degradation. This issue, which persists across existing methods, remains unresolved. We attribute it to the loss of texture information during the illumination recovery process. To address this limitation in future research, we propose extracting texture priors from other domains, such as the frequency domain (Xu et al., 2022; He et al., 2019). These priors could complement the reflectance priors extracted from the RGB domain, thereby improving the preservation of critical texture features.

Additionally, we aim to combine our method with more cutting-edge deep priors, such as that from segment anything model (He et al., 2024), depth anything model (Chen et al., 2024a), or other large foundation model (Tang et al., 2024), and extend our method to wider applications, such as image super-resolution (Chen et al., 2023d; 2024b), deblurr (Chen et al., 2023e; Zhao et al., 2023a), desnow (Chen et al., 2023c;b), dehaze (Fang et al., 2024a; Zhao et al., 2024), derain (Chen et al., 2023a; Zhao et al., 2023b), conditional fusion (He et al., 2023; Xu et al., 2023; Ju et al., 2022). Combining our strategies with cutting-edge algorithms, such as spiking neural network (Wang et al., 2023; Fang et al., 2024b) and mamba (Xiao et al., 2025; 2024b), is also expected to bring better performance.

Besides, enhanced methods are also expected to facilitate downstream tasks, such as image segmentation (Xiao et al., 2024a; 2023) and detection (He et al., 2025; Pu et al., 2024; 2023). To achieve this, several strategies are expected to be integrated into our method, including generating downstream-friendly data (Yuan et al., 2024a;b), designing specific augmentation strategies (Ma et al., 2024a;b;c).

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
