# OpenReview forum: "Reti-Diff: Illumination Degradation Image Restoration with Retinex-based Latent Diffusion Model"
_ICLR.cc/2025/Conference — ICLR 2025 Spotlight_

### Official Review · Reviewer_72tr · 2024-10-26

**Soundness:** 3
**Presentation:** 4
**Contribution:** 3
**Rating:** 8
**Confidence:** 5

**Summary:**

This paper focuses on designing a latent diffusion model that utilizes the Retinex priors to address the issue of illumination degradation. To be detailed, the authors developed a module for estimating the Retinex priors from the low-quality and high-quality images and a Transformer backbone for the image enhancement. These components work in synergy to restore images. The authors provide extensive experiments to demonstrate the effectiveness and advancement of their method.

**Strengths:**

1. The coupling of latent diffusion model with the Retinex priors and Transformer is intriguing. And exploring the robustness of network with the Retinex theory in illumination degradation scenarios is meaningful.
2. The proposed method, compared to existing diffusion models in image enhancement, demonstrates relatively high computational efficiency and lower resource consumption while maintaining strong performance.
3. The experimental validation in the paper is thorough, particularly regarding downstream tasks and evaluations across multiple datasets.
4. The paper is well-organized, clearly written, and easy to understand.

**Weaknesses:**

1. The paper utilizes diffusion models to get the latent Retinex priors, but further analysis is needed to explore the relationship between diffusion models and the designed latent Retinex priors. (insights)
2. The mechanisms by which the Retinex priors operate need to be clarified. Ensuring the consistency of the Retinex theory in the latent space is a question worthy of investigation. Further deepening the results-oriented experimental conclusions in the mechanistic level would improve the quality of study.
3. The design of the method results in a significant reliance on the modeling of Retinex theory. However, there seems to be little discussion on the reliability of Retinex decomposition.

**Questions:**

1. What’s the motivation of using the latent diffusion model to generate the Retinex priors? What are its advantages? Would it be more efficient and reliable to use a single Transformer or CNN network for it?
2. DiffIR [1] achieves image restoration using a similar prior. What would be the difference if the entire image enhancement process were simply treated as the prior, compared to using the Retinex-based model as the priors? Demonstrating the necessity of the latent Retinex priors would be more valuable.
3. About the Retinex priors, how does it perform in terms of prior feature errors when exchanging features between the ground truth and low-light images? In addition to the qualitative and quantitative evaluations of the enhancement results, it would be beneficial to conduct an assessment at the prior level as well.
4. It is recommended to perform a convergence analysis of using the latent Retinex priors. For example, showcasing the changes in loss or evaluation metrics on the training and validation sets would be beneficial.
5. Given the introduction of the Retinex priors, does the proposed method heavily depend on the reliability of Retinex decomposition? Since this is an ill-posed problem, it would be helpful to provide an explanation or at least a few qualitative results of the Retinex decomposition to support this claim.
6. How does the method perform in extreme conditions, specifically when the Retinex decomposition fails? (Extremely bright light and strong shadows)

Although there are some concerns, they do not detract from the fact that this is a relatively well-researched paper. I am inclined to provide a positive opinion in this stage.

[1] Xia, Bin, et al. "Diffir: Efficient diffusion model for image restoration." Proceedings of the IEEE/CVF International Conference on Computer Vision. 2023.

---

> ### Author Response · Authors · 2024-11-19
> **Rebuttal by authors (1/3)**
>
> **W1&Q1: Why use LDM rather than Transformer or CNN?**
>
> We introduce an LDM to extract compact Retinex priors, aiming to directly estimate high-fidelity priors from low-quality input images. Using an LDM enables more accurate prior estimation compared to other structures, such as CNNs, thereby enhancing the robustness and generalizability of image enhancement in extreme degradation scenarios. To validate this, we replaced our LDM prior estimator with alternative structures of similar computational cost and found that our LDM achieves superior performance.
>
> |||LDM (Ours)|CNN|Transformer|Linear|
> |---|---|---|---|---|---|
> |LOLv2-r|PSNR|**22.97**|21.22|21.81|21.72|
> ||SSIM|**0.858**|0.851|0.846|0.830|
> |LOLv2-s|PSNR|**27.53**|27.08|27.27|25.83|
> ||SSIM|**0.951**|0.951|0.950|0.927|
>
> **W2: Clarifying the Retinex operation mechanisms and more experiments.**
>
> * Our RGformer uses cross-attention to model the Retinex theory with the purpose of refining and aggregating features modulated by reflectance and illumination priors, ensuring enhancement performance. To achieve this, we originally employed two different attention mechanisms to model the Retinex theory. The first mechanism denoted as A1, involves setting  $\mathbf{Q}=\mathbf{W} _ Q \mathbf{F} _ {\mathbf{R}}$, $\mathbf{K}=\mathbf{W} _ K \mathbf{F} _ {\mathbf{R}}$, and $\mathbf{V}=\mathbf{W} _ V \mathbf{F} _ {\mathbf{L}}$. The second mechanism, termed A2 (Ours), is a cross-attention. The multiplication of $\mathbf{Q}\mathbf{K}$ is initially performed as $\widetilde{\text{H}}\widetilde{\text{W}}\times\widetilde{\text{C}}$ @ $\widetilde{\text{C}}\times\widetilde{\text{H}}\widetilde{\text{W}}$ in an element-wise manner, and the resulting coefficient matrix with a size of $\widetilde{\text{H}}\widetilde{\text{W}}\times\widetilde{\text{H}}\widetilde{\text{W}}$ is then element-wise multiplied with $\mathbf{V}$. In this context, A1 explicitly models the Retinex theory, while A2 does so implicitly. As shown in Table, A2 achieves better results, suggesting the effect of the implicit model.
> * As demonstrated in **Fig. 7**, the breakdown ablation and swap experiments confirm that our Retinex prior effectively guides the enhancement process in both illumination restoration and detail enhancement. These results validate that our method successfully models Retinex theory within the latent space.
>
> |||A1|A2 (Ours)|
> |---|---|---|---|
> |LOLv2-r|PSNR|22.06|**22.97**|
> ||SSIM|0.844|**0.858**|
> |LOLv2-s|PSNR|26.44|**27.53**|
> ||SSIM|0.949|**0.951**|
>
> **W3: The reliability of Retinex decomposition**
>
> * In the implementation, the decomposition network $D(\cdot)$ consists of three Conv+LeakyReLU layers followed by a Conv+ReLU layer, utilizing the pretrained model released by URetinex [1]. After obtaining the Retinex components from the low-quality input, RLDM is employed to estimate high-quality Retinex priors from these low-quality components, which are subsequently used to guide the enhancement process.
>
> * For fairness, we do not selectively optimize the decomposition model. However, it is evident that a more robust decomposition model yields superior Retinex components, thereby enhancing overall performance. Additionally, the effectiveness of both the prior extraction manner and the prior incorporation strategy has been validated in **Tables 6 and 7**.
>
> * Beyond these evaluations, we further demonstrate in **Table 7 and Fig. 7** that our method maintains strong performance even when the Retinex priors are partially compromised due to extreme environmental conditions. This comprehensive analysis and experimental evidence affirm the reliability and robustness of our method to the Retinex theory.
>
> **Q2. Why use Retinex priors rather than the RGB prior?**
>
> We propose extracting Retinex priors to guide the enhancement process through a decomposition approach, which is tailored to the IDIR task. We contend that relying solely on priors extracted from the RGB domain struggles to fully represent valuable texture details and correct illumination cues, leading to suboptimal restoration performance. To verify this, we substitute our RLDM for the LDM structure used in DiffIR. In *LOL-v2-syn*, we observe that the PSNR rises from 24.76 to 26.14 and the SSIM increases from 0.921 to 0.933.
>
> [1] URetinex, CVPR 2022.

---

> ### Author Response · Authors · 2024-11-19
> **Rebuttal by authors (2/3)**
>
> **Q3. Difference between low-quality image and GT at the prior level**
>
> We further utilize the Mean Squared Error (MSE) metric to quantify the differences between the Retinex priors extracted from low-quality images and those derived from the ground truth. As shown in the table, the observed differences are minimal, underscoring the effectiveness of our method in directly extracting high-quality priors from degraded inputs. Additionally, to validate that our extracted priors are more aligned with those obtained from the ground truth, we introduce two baselines: **$\mathbf{Z} _ \mathbf{R}$-Ablation1** and **$\mathbf{Z} _ \mathbf{L}$-Ablation2**. **$\mathbf{Z} _ \mathbf{R}$-Ablation1** represents the MSE between $\mathbf{Z} _ \mathbf{R}$ extracted from the ground truth and the ablated version of "w/ $\mathbf{Z} _ \mathbf{R}$" in **Table 7**, while **$\mathbf{Z} _ \mathbf{L}$-Ablation2** denotes the MSE between $\mathbf{Z} _ \mathbf{L}$ from the ground truth and the ablated version of "w/ $\mathbf{Z} _ \mathbf{L}$" in **Table 7**. The results verify that our Reti-Diff can extract high-quality priors directly from the low-quality inputs.
>
> |MSE|LOLv1|LOLv2-r|LOLv2-s|
> |---|---|---|---|
> |$\mathbf{Z} _ \mathbf{R}$-Ours|**0.00211**|**0.00317**|**0.00181**|
> |$\mathbf{Z} _ \mathbf{L}$-Ours|**0.00429**|**0.00691**|**0.00261**|
> |$\mathbf{Z} _ \mathbf{R}$-Ablation1|0.00375|0.00436|0.00332|
> |$\mathbf{Z} _ \mathbf{L}$-Ablation2|0.00608|0.00783|0.00417|
>
> **Q4. Convergence analysis**
>
> We perform ablation experiments to evaluate convergence across different priors. After confirming network convergence, we record the average batch loss over the subsequent ten epochs under identical training conditions, with the results summarized in the table. The findings reveal that our method, which integrates the Retinex prior, achieves a lower loss compared to approaches utilizing the RGB prior or no prior. These results underscore the superior convergence performance of our Reti-Diff model when guided by the Retinex prior. Notably, as both the image prior and Retinex prior employ customized loss functions, the comparison is restricted to the overlapping components of these functions to ensure fairness.
>
> ||w/oprior|RGB prior|Retinex prior (Ours)|
> |---|---|---|---|
> |LOLv2-r|6.20186e-01|4.93128e-01|**4.12497e-01**|
> |LOLv2-s|4.84951e-01|4.48667e-01|**3.17285e-01**|
>
> **Q5. Visualization and analysis of Retinex decomposition**
>
> * We provide a visualization of the Retinex components of the low-quality input and the ground truth decomposed by D(⋅) in **Fig. S6**. As shown, the Retinex components extracted from the low-quality input exhibit significant degradation compared to those from the ground truth. However, as demonstrated in the table provided in response to Q3, our Retinex priors show minimal differences from the ground truth priors. This finding confirms that our method does not heavily depend on the quality of the Retinex decomposition.
>
> * Furthermore, as illustrated in **Table 7 and Fig. 7**, swapping our Retinex priors with those derived from the ground truth results in only marginal improvements. This observation underscores that our Reti-Diff framework is capable of directly extracting high-quality priors from low-quality images.

---

> ### Author Response · Authors · 2024-11-19
> **Rebuttal by authors (3/3)**
>
> **Q6. Performance in extreme conditions**
>
> Three extreme cases can lead to suboptimal Retinex decomposition: (1) **Severe environmental degradation** (blurred texture details), (2) **Extreme lighting conditions** (uneven lighting), and (3)
> Situations where the **Retinex components of the degraded image are nearly identical to those of the ground truth**.
>
> * In the first extreme case, our method addresses this by extracting highly compact Retinex priors from low-quality inputs, enabling robust enhancement despite complex degradation. To validate its effectiveness, we assess performance on the underwater image enhancement (UIE) task, which inherently suffers from noise, color casts, and distortion. As shown in the table, our method achieves state-of-the-art results. Additionally, we employ the advanced image quality assessment method DA-CLIP [2] to identify the 15 underwater images with the lowest quality and most severe illumination issues, forming two sub-datasets: LSUI_Extreme and UIEB_Extreme. Performance on these sub-datasets further underscores our method's leading capability to handle extreme degradation scenarios.
>
> * In the second extreme case, despite interference from highly imbalanced or inconsistent lighting, our Retinex priors effectively capture valuable information from degraded inputs, providing stable guidance for enhancement. This is demonstrated by our superior performance in the backlit image enhancement task, where images suffer from significant illumination imbalance. By analyzing pixel distribution histograms, we select the 15 most challenging images with extreme and complex lighting, creating the BAID_Extreme dataset. As shown in the table, our method continues to outperform competitors, achieving leading results even under such demanding conditions.
>
> * In the third extreme case, when the Retinex components of the degraded image closely resemble those of the ground truth, our method remains effective. Results in **Table 7 and Fig. S1** highlight the superiority of our Retinex priors in guiding the enhancement process, ensuring robust performance even in these challenging cases.
>
>
> |ExtremeCase I||U-Shape|AST|MambaIR|Ours|
> |---|---|---|---|---|---|
> |LSUI|PSNR|24.16|27.46|27.68|**28.10**|
> ||SSIM|0.917|0.916|0.916|**0.929**|
> |UIEB|PSNR|22.91|22.19|22.60|**24.12**|
> ||SSIM|0.905|0.908|0.939|**0.910**|
> |LSUI_Extreme|PSNR|19.76|20.37|20.02|**21.82**|
> ||SSIM|0.786|0.786|0.787|**0.789**|
> |UIEB_Extreme|PSNR|17.79|17.59|18.41|**19.77**|
> ||SSIM|0.817|0.852|0.872|**0.878**|
>
> |ExtremeCase II||CLIP-LIT|AST|MambaIR|Ours|
> |---|---|---|---|---|---|
> |BAID|PSNR|21.13|22.61|23.07|**23.19**|
> ||SSIM|0.853|0.851|0.874|**0.876**|
> |BAID_Extreme|PSNR|20.06|22.65|22.95|**23.03**|
> ||SSIM|0.653|0.761|0.757|**0.776**|
>
> [2] DA-CLIP, ICLR 2024.

---

> > ### Comment · Reviewer_72tr · 2024-11-21
> >
> > The authors' responses have resolved most of my concerns. Therefore, I am inclined to keep my original score.

---

> > > ### Author Response · Authors · 2024-11-21
> > > **Thanks for recognizing the value of our work!**
> > >
> > > We wish to express our sincere appreciation to the reviewer for recognizing the substantial significance of our Reti-Diff's contribution, specifically the plug-and-play Retinex prior estimator, RLDM, and the prior-guided image restoration network, RGformer.
> > >
> > > Your acknowledgment holds great importance to us and serves as a meaningful validation of our dedicated efforts to advance this critical area of research.

---

### Official Review · Reviewer_NsRN · 2024-11-04

**Soundness:** 3
**Presentation:** 3
**Contribution:** 3
**Rating:** 8
**Confidence:** 5

**Summary:**

This paper introduces the Retinex-based LDM solution, Reti-Diff, for illumination degradation image restoration (IDIR) tasks, aiming to generate visual fidelity results while decreasing computational burdens. In specific, Reti-Diff proposes RLDM to acquire Retinex priors and then leverage RGformer to guide the image restoration process with the extracted priors, resulting in the production of refined images with consistent content and robustness to handle complex degradation scenarios. Extensive experiments demonstrate that Reti-Diff outperforms existing methods on three IDIR tasks, as well as downstream applications.

**Strengths:**

1. This paper proposes RLDM to extract Retinex priors, and RGformer to integrate the priors, ensuring robustness and generalization in complex illumination degradation scenarios.
2. This paper is well organized.
3. Experiments on four IDIR tasks verify the superiority, efficiency, and generalizability of the method and demonstrate that Retinex priors can serve as a plug-and-play strategy to improve the quality of existing methods.

**Weaknesses:**

1. More related works are expected to be discussed, for example, low-light image enhancement via clip-fourier guided wavelet diffusion[1].
2. Why do the authors choose to use a cross-attention mechanism to model the Retinex theory?
3. The authors are encouraged to highlight the motivation for extracting Retinex priors and explain why using a diffusion model to extract the priors.
4. Additionally, in the experiment, the authors are expected to verify if the Retinex priors perform better than the sole RGB prior and if the diffusion model can serve as a better predictor than the common network structure.
5. In Phase 2, the authors first train the RLDM before conducting joint training with RGFormer. In the ablation study, the authors provide results without the joint training stage, but they do not compare the results of removing the independent RLDM training stage. Would it yield better results if RLDM, RPE, and RGFormer were trained directly together?
6. Some typo errors are expected to be fixed, for example, ZR and ZL in Line 90 in the Supp.

[1] Xue M, He J, He Y, et al. Low-light image enhancement via clip-fourier guided wavelet diffusion[J]. arXiv preprint arXiv:2401.03788, 2024.

**Questions:**

Can the authors analyze the reason for the failure cases? Will this be because of the failure of one of the Retinex priors or because of the failure of RGformer to integrate the priors?

---

> ### Author Response · Authors · 2024-11-19
> **Rebuttal by authors (1/2)**
>
> **W1. Lacking related works**
>
> The recommended method [1], along with two other state-of-the-art approaches [2, 3], is discussed in further detail under "Related Work".
>
> **W2. Why use cross-attention?**
>
> Our RGformer uses cross-attention to model the Retinex theory with the purpose of refining and aggregating features modulated by reflectance and illumination priors, ensuring enhancement performance. To achieve this, we originally employed two different attention mechanisms to model the Retinex theory. The first mechanism denoted as A1, involves setting  $\mathbf{Q}=\mathbf{W} _ Q \mathbf{F} _ {\mathbf{R}}$, $\mathbf{K}=\mathbf{W} _ K \mathbf{F} _ {\mathbf{R}}$, and $\mathbf{V}=\mathbf{W} _ V \mathbf{F} _ {\mathbf{L}}$. The second mechanism, termed A2 (Ours), is a cross-attention. The multiplication of $\mathbf{Q}\mathbf{K}$ is initially performed as $\widetilde{\text{H}}\widetilde{\text{W}}\times\widetilde{\text{C}}$ @ $\widetilde{\text{C}}\times\widetilde{\text{H}}\widetilde{\text{W}}$ in an element-wise manner, and the resulting coefficient matrix with a size of $\widetilde{\text{H}}\widetilde{\text{W}}\times\widetilde{\text{H}}\widetilde{\text{W}}$ is then element-wise multiplied with $\mathbf{V}$. In this context, A1 explicitly models the Retinex theory, while A2 does so implicitly. As shown in Table, A2 achieves better results, suggesting the effect of the implicit model.
>
> |||A1|A2 (Ours)|
> |---|---|---|---|
> |LOLv2-r|PSNR|22.06|**22.97**|
> ||SSIM|0.844|**0.858**|
> |LOLv2-s|PSNR|26.44|**27.53**|
> ||SSIM|0.949|**0.951**|
>
> **W3. Why extract Retinex priors and employ a diffusion model to extract them?**
>
> * We propose extracting Retinex priors to guide the enhancement process through a decomposition approach. By separating features into reflectance and illumination components under the guidance of Retinex priors, our method effectively addresses texture loss and illumination distortion, enabling the generation of enhanced images with coherent content.
>
> * We introduce a latent diffusion model (LDM) to extract compact Retinex priors, aiming to directly estimate high-fidelity priors from low-quality input images. Using an LDM enables more accurate prior estimation compared to other structures, such as CNNs, thereby enhancing the robustness and generalizability of image enhancement in extreme degradation scenarios. To validate this, we replaced our LDM prior predictor with alternative structures of similar computational cost and found that our LDM achieves superior performance.
>
> |||CNN|Transformer|Linear|LDM (Ours)|
> |---|---|---|---|---|---|
> |LOLv2-r|PSNR|21.22|21.81|21.72|**22.97**|
> ||SSIM|0.851|0.846|0.830|**0.858**|
> |LOLv2-s|PSNR|27.08|27.27|25.83|**27.53**|
> ||SSIM|0.951|0.950|0.927|**0.951**|
>
> **W4. Comparisons of different priors and different prior extraction strategies**
>
> * We propose extracting Retinex priors to guide the enhancement process through a decomposition approach, which is tailored to the IDIR task. We contend that relying solely on priors extracted from the RGB domain struggles to fully represent valuable texture details and correct illumination cues, leading to suboptimal restoration performance.
>
> * We provide comprehensive evidence that our Retinex priors facilitate superior image enhancement compared to RGB priors across various conditions, including standard IDIR settings and extreme IDIR scenarios. In the standard setting, as presented in **Table 1**, we confirm that Reti-Diff with Retinex priors achieves an average performance gain of 20.6% over the state-of-the-art DiffIR method [4], which employs only RGB priors. Additionally, our ablation study (**Table 7**) reveals that our approach with Retinex priors (denoted as w/ $\mathbf{Z}$) consistently outperforms our method with RGB priors alone (denoted as w/o $\mathbf{Z}$). Similar trends are observed under two extreme IDIR conditions in **Table 7**.
>
> * Furthermore, we demonstrate that our LDM serves as a more effective predictor than conventional network architectures, as highlighted in response to W3.
>
> |||RGB prior|Retinex prior (Ours)|
> |---|---|---|---|
> |LOLv2-r|PSNR|21.63|**22.97**|
> ||SSIM|0.830|**0.858**|
> |LOLv2-s|PSNR|26.25|**27.53**|
> ||SSIM|0.939|**0.951**|
>
> **W5. Effect of pretrain**
>
> We pretrain the RLDM to ensure robust optimization of the latent diffusion model and employ a joint training strategy to enhance restoration quality further. When the pretraining phase is omitted, we observe a decline in performance, underscoring its importance in achieving optimal results.
>
> |||w/ pretrain (Ours)|w/o pretrain|
> |---|---|---|---|
> |LOLv2-r|PSNR|**22.97**|22.09|
> ||SSIM|**0.858**|0.851|
> |LOLv2-s|PSNR|**27.53**|26.90|
> ||SSIM|**0.951**|0.949|
>
>
> **W6. Typo errors**
>
> We have carefully revised the paper and fixed the typos.
>
>
>
>
> [1] CFWD, arXiv 2024.
>
> [2] UNIE, ECCV 2022.
>
> [3] NTD, ACM MM 2023.
>
> [4] DiffIR, ICCV 2023.

---

> ### Author Response · Authors · 2024-11-19
> **Rebuttal by authors (2/2)**
>
> **Q1: The reasons for the failure cases**
>
> The reasons for the failure cases are discussed in Section E in the supplementary material. Here we briefly explain the reasons.
>
> In the first image of **Fig. S11**, our method fails to distinguish the two regions marked by the dashed box due to the ambiguous boundary of the right region and the intrinsic similarities shared between the two areas. Consequently, our method interprets the two distinct regions as a single object and attempts to merge them. This behavior contrasts with the successful separation of the lower clothing, which exhibits more apparent differences. A similar issue is observed in the second image.
>
> This limitation is not attributable to the Retinex priors or the RGformer, as neither is specifically designed to highlight subtle differences. To address this challenge, future research could explore extracting texture priors from alternative domains, such as the frequency domain. Such priors could complement those in the RGB domain by emphasizing subtle distinctions.

---

> > ### Comment · Reviewer_NsRN · 2024-11-24
> >
> > My concerns have been addressed. I think it is a good work and recommend acceptance.

---

> > > ### Author Response · Authors · 2024-11-24
> > > **Thanks for recognizing the value of our work!**
> > >
> > > We wish to express our sincere appreciation to the reviewer for recognizing the substantial significance of our Reti-Diff's contribution, specifically the acknowledgment of RLDM and RGformer for their robustness and generalization in complex illumination degradation scenarios.
> > >
> > > Your acknowledgement holds great importance to us and serves as a meaningful validation of our dedicated efforts to advance this critical area of research.

---

### Official Review · Reviewer_uVsX · 2024-11-04

**Soundness:** 3
**Presentation:** 3
**Contribution:** 3
**Rating:** 8
**Confidence:** 5

**Summary:**

The paper introduces Reti-Diff, a novel latent diffusion model-based framework tailored for Illumination Degradation Image Restoration (IDIR). It leverages Retinex theory to improve image quality by using two primary components: the Retinex-based Latent Diffusion Model (RLDM) and the Retinex-guided Transformer (RGformer).

**Strengths:**

1. The paper combined the Retinex model and data-driven methods since the first effort of RetinexNet.
2. Experiments are sufficient.
3. Code is a plus.

**Weaknesses:**

1. For the low-light enhancement task, I want to see more results on LIME, NPE, MEF, DICM and VV. Since these datasets do not have ground truth, then is more fair to justify the effectiveness. Any visual results?

2. Missing citations of real-world low-light enhancement methods,
[1] Unsupervised Night Image Enhancement: When Layer Decomposition Meets Light-Effects Suppression
[2] Enhancing Visibility in Nighttime Haze Images Using Guided APSF and Gradient Adaptive Convolution

3. LDM is slow in inference.
The paper lacks a discussion on the computational complexity and runtime efficiency of the proposed model.

**Questions:**

1. How does the proposed method handle edge cases such as extreme noise or heavy color distortions, which may not follow typical low-light degradation patterns?

2. Is there a recommended strategy for tuning the noise variance parameters in the latent diffusion process for optimal performance across varied image qualities?

3. How does Reti-Diff perform in scenarios with extreme lighting conditions, such as overexposed or underexposed images, which may challenge the reliability of Retinex-based priors?

---

> ### Author Response · Authors · 2024-11-19
> **Rebuttal by authors**
>
> **W1: More results in real-world datasets**
>
> * We present the visualization results of the real-world IDIR task in **Fig. S5**, where our method demonstrates superior performance in refining texture details and correcting inconsistent illumination, even under complex real-world degradation conditions.
>
> * Additionally, we provide an expanded evaluation with three additional metrics, comparing our model against two state-of-the-art methods. As shown in the table, our approach consistently outperforms the alternatives, securing a leading position in the comparison.
>
> |LOLv2Syn||DICM|LIME|MEF|NPE|VV|
> |-|-|-|-|-|-|-|
> |GSAD|BRISQUE↓|13.953|17.694|16.612|20.292|14.835|
> ||NIMA↑|5.196|4.799|4.848|4.924|3.909|
> ||MUSIQ↑|60.094|56.869|57.741|59.369|42.779|
> |PairLIE|BRISQUE↓|29.664|25.212|33.600|30.992|35.877|
> ||NIMA↑|4.269|4.391|4.609|4.303|3.957|
> ||MUSIQ↑|59.577|59.411|59.499|61.361|38.187|
> |Ours|BRISQUE↓|**8.435**|**12.033**|**15.345**|**15.703**|**14.600**|
> ||NIMA↑|**5.234**|**5.138**|**4.870**|**5.054**|**3.997**|
> ||MUSIQ↑|**67.129**|**63.034**|**61.171**|**66.722**|**44.831**|
>
> **W2: Missing citations**
>
> We have discussed the two recommended and relevant papers in the "Related Work".
>
> **W3: Computational complexity & runtime**
>
> As shown in **Fig. 8**, our method overcomes the inherent low-speed limitation of diffusion models by utilizing the LDM to estimate highly compact Retinex priors. This approach requires much fewer iterations, specifically set to 4, than other DM-based IDIR solutions. **Table 2** further verifies the efficiency of our method, with comparisons of **Parameters**, **MACs**, and **FPS** demonstrating its superiority in terms of storage efficiency, computational cost, and runtime performance.
>
> **Q1: How to handle extremely degraded cases?**
>
> Our method extracts highly compact priors from low-quality inputs to guide the enhancement process, demonstrating resilience against complex degradation. To validate this, we evaluate our performance on the underwater image enhancement (UIE) task, which is inherently affected by noise, color cast, and distortion. As shown in the table, our method achieves a leading position. Furthermore, we utilize the state-of-the-art image quality assessment method, DA-CLIP [1], to identify the 15 underwater images with the lowest quality and most severe illumination issues, representing the most extreme degradation cases. These form two sub-datasets, LSUI_Extreme and UIEB_Extreme. As evidenced in the table, our method consistently outperforms others, achieving the best performance.
>
> |||U-Shape|AST|MambaIR|Ours|
> |-|-|-|-|-|-|
> |LSUI|PSNR|24.16|27.46|27.68|**28.10**|
> ||SSIM|0.917|0.916|0.916|**0.929**|
> |UIEB|PSNR|22.91|22.19|22.60|**24.12**|
> ||SSIM|0.905|0.908|0.939|**0.910**|
> |LSUI_Extreme|PSNR|19.76|20.37|20.02|**21.82**|
> ||SSIM|0.786|0.786|0.787|**0.789**|
> |UIEB_Extreme|PSNR|17.79|17.59|18.41|**19.77**|
> ||SSIM|0.817|0.852|0.872|**0.878**|
>
> **Q2: Strategy for tuning the noise variance**
>
> * For fairness, we adopt the default parameter optimization strategy used by DiffIR [2]. As shown in **Tables 2–5** of our experiments, Reti-Diff demonstrates the ability to extract valuable priors, ensuring high-quality image restoration across four IDIR tasks.
>
> * We acknowledge that further refining the optimization strategy could enhance the diffusion model's capacity to address noise across varying levels and facilitate the generation of higher-quality enhanced images [3, 4]. We have included added this as a future research direction and will thoroughly investigate how this strategy can contribute to developing a more effective IDIR algorithm.
>
> **Q3: Performance in extreme lighting conditions**
>
> * Our proposed Retinex prior effectively captures valuable information from the Retinex components extracted from the original degraded input. Even when these components are inadequately captured due to interference from extreme lighting conditions, the compact Retinex priors compensate for this limitation, guiding the decomposition of image features. This approach facilitates the generation of refined images with consistent content and enhances robustness in handling complex degradation scenarios.
>
> * Our method demonstrates superior performance in backlit image enhancement, effectively addressing challenges posed by inconsistent lighting conditions. Besides, we select the top 15 images with the most inconsistent illumination by analyzing their pixel distribution histograms and thus form a dataset with extreme and complex lighting conditions, BAID_Extreme. As shown in the table, our method still achieves a leading place in this task.
>
> |||CLIP-LIT|AST|MambaIR|Ours|
> |-|-|-|-|-|-|
> |BAID|PSNR|21.13|22.61|23.07|**23.19**|
> ||SSIM|0.853|0.851|0.874|**0.876**|
> |BAID_Extreme|PSNR|20.06|22.65|22.95|**23.03**|
> ||SSIM|0.653|0.761|0.757|**0.776**|
>
> [1] DA-CLIP, ICLR 2024.
>
> [2] DiffIR, ICCV 2023.
>
> [3] Common Diffusion Noise Schedules and Sample Steps are Flawed, WACV 2024.
>
> [4] MuLAN, NeurIPS 2024.

---

> > ### Comment · Reviewer_uVsX · 2024-11-20
> > **all my concerns addressed**
> >
> > The authors address all my concerns. I raised my score to 8: Accept, good paper.

---

> > > ### Author Response · Authors · 2024-11-20
> > > **Thanks for recognizing the value of our work!**
> > >
> > > We wish to express our sincere appreciation to the reviewer for recognizing the substantial significance of our Reti-Diff's contribution, specifically the plug-and-play Retinex prior estimator, RLDM, and the prior-guided image restoration network, RGformer.
> > >
> > > Your acknowledgement holds great importance to us and serves as a meaningful validation of our dedicated efforts to advance this critical area of research.

---

### Official Review · Reviewer_p8Wx · 2024-11-07

**Soundness:** 3
**Presentation:** 3
**Contribution:** 2
**Rating:** 6
**Confidence:** 5

**Summary:**

The paper proposes to use diffusion models in conjunction with a transformer model to perform image restoration of poorly lit images. The proposed method is evaluated in 4 different settings and shown the merits.

**Strengths:**

+ The paper looks at an important problem of illumination restoration in images which is quite relevant for downstream tasks.

**Weaknesses:**

- The use of intrinsic image decomposition through Retinex is a dated idea. There are several better methods available that have neither been explored nor used. https://www.elenagarces.es/projects/SurveyIntrinsicImages/

- The work lacks novelty as it uses diffusion to decompose the image and a transformer to reconstruct it. The method has hardly any contribution.

- Though the evaluation of the method is done for several conditions, the process is not built correctly with any conviction, which is a downer.

- The method uses a couple of downstream applications to demonstrate the merits: detection and segmentation. The tasks are very easy to generalize, given the power of deep networks without illumination enhancement. The work could not convince the reader about the merit of this approach.

**Questions:**

See above.

---

> ### Author Response · Authors · 2024-11-19
> **Rebuttal by authors （1/2）**
>
> **W1: Why choose Retinex theory rather than other better models?**
>
> * We adopt Retinex theory due to its notable contributions to the field of Illumination Degradation Image Restoration (IDIR). According to recent estimates, over 100 papers in the last three years have leveraged Retinex theory to address IDIR problems, underscoring its prominence in this domain. Ranging from traditional methods [1,2] to advanced architectures like convolutional neural networks [3,4], transformers [5,6], and diffusion models [7-9], the Retinex prior has consistently played an essential role in addressing IDIR challenges and achieving state-of-the-art results.
>
> * Motivation: Different from existing practices of Retinex theory, we first propose to use a latent diffusion model to directly estimate Retinex priors from the low-quality inputs and then use the priors to guide a transformer to enhance degraded images, ensuring texture enhancement and illumination restoration. For its strong generalization, this framework can even function as a plug-and-play method to improve the performance of existing methods by using Retinex priors to guide their enhancement process (See **Table 8**).
>
> * We evaluate another intrinsic prior based on the Simplified Lambertian Intrinsic Model (SLIM), introduced in TOG 2024 [10], and construct a comparative method termed SLIM by directly replacing our Retinex priors with SLIM priors. Notably, the decomposition network in SLIM is highly complex, comprising 337M parameters, whereas our framework employs a simpler UNet-based decomposition network. As shown in the table, our framework utilizing Retinex priors consistently outperforms the one using SLIM priors, further validating the superiority of our proposed Reti-Diff.
>
> |||SLIM|Retinex Priors (Ours)|
> |-|-|-|-|
> |LOLv2-r|PSNR|22.02|**22.97**|
> ||SSIM|0.840|**0.858**|
> |LOLv2-s|PSNR|26.107|**27.53**|
> ||SSIM|0.936|**0.951**|
> |LOLv1|PSNR|24.18|**25.35**|
> ||SSIM|0.853|**0.866**|
> |LSUI|PSNR|27.51|**28.10**|
> ||SSIM|0.902|**0.929**|
> |UIEB|PSNR|22.52|**24.12**|
> ||SSIM|0.900|**0.910**|
> |BAID|PSNR|22.57|**23.19**|
> ||SSIM|0.859|**0.876**|
>
> **W2: Lack of novelty for simply combining diffusion model and transformer**
>
> * Our approach represents a **pioneering effort** in introducing a latent diffusion model to address the IDIR problem. The **real working mechanism** of Reti-Diff involves employing a latent diffusion model (RLDM) to estimate Retinex priors from low-quality inputs, which subsequently guides the transformer (RGformer) in enhancing those inputs.
>
> * Our proposed RLDM functions as a plug-and-play module that can be seamlessly integrated into various frameworks. As presented in **Table 8**, our RLDM can further improve the performance of existing methods. This result highlights the strong generalizability of our Retinex priors and underscores that our framework is not a mere combination of diffusion model and transformer but a novel and effective solution. Importantly, because RLDM estimates vector-level Retinex priors rather than directly restoring degraded images, the computational cost of our method is substantially lower than other diffusion model-based IDIR solutions (see **Table 2**).
>
> * The carefully tailored design of Reti-Diff enables it to restore degraded images with enhanced details and balanced illumination, even in challenging degradation scenarios. Extensive experiments across four IDIR tasks and three downstream applications consistently highlight the superior performance and efficiency of our approach.
>
> **W3: Incorrect evaluation across different conditions**
>
> * In total, our experiments abandon the GT-mean strategy in all experiments and only compare with those methods abandoning the GT-mean for a fair comparison.
>
> * In specific tasks, we follow the corresponding common practice, including the use of training and testing sets and the selection of the corresponding cutting-edge methods. For instance, we follow the methodology of Retformer [5] for low-light image enhancement and NU2Net [11] for underwater image enhancement. Similarly, we adopt the training approach of CLIP-LIT [12] for backlit image enhancement and CIDNet [13] for real-world illumination degradation image restoration. These practices ensure that our experiments are conducted fairly and provide reliable comparisons.
>
> * If any experiments remain unclear, please do not hesitate to reach out. We are very happy to clarify and address any misunderstandings.
>
> [1] A weighted variational model for simultaneous reflectance and illumination estimation. CVPR 2016
>
> [2] Structure-revealing low-light image enhancement via robust retinex model. TIP 2018
>
> [3] Uretinex-net. CVPR 2022
>
> [4] CRetinex. IJCV 2024
>
> [5] Retformer. ICCV 2023
>
> [6] RFR. CVPR 2023
>
> [7] Diff-retinex. ICCV 2023
>
> [8] Retinex-diffusion. arXiv 2024
>
> [9] DI-Retinex. arXiv 2024
>
> [10] Colorful Diffuse Intrinsic Image Decomposition in the Wild, TOG 2024
>
> [11] NU2Net, AAAI 2023
>
> [12] CLIP-LIT, ICCV 2023
>
> [13] CIDNet, arXiv 2024

---

> > ### Author Response · Authors · 2024-11-19
> > **Rebuttal by authors （2/2)**
> >
> > **W4: The detection and segmentation of low-quality images can be easily addressed by deep networks.**
> >
> > As indicated in the "Baseline" rows of **Tables 10, 11, and 12**, our experiments demonstrate that state-of-the-art detectors and segmenters face significant challenges when processing low-quality images without enhancement. However, when these images are enhanced—especially using our proposed Reti-Diff—the performance of detection and segmentation improves substantially. Specifically, Reti-Diff achieves performance gains of 14.2% in mAP for low-light image detection, 22.2% in mIoU for low-light semantic segmentation, and 32.9% in the $M$ metric for low-light concealed object segmentation tasks.

---

> ### Author Response · Authors · 2024-11-23
> **Thanks for recognizing the value of our work and updating the score to 6!**
>
> We notice that Reviewer p8Wx has updated the score to 6, indicating that our rebuttal has played a positive role in resolving the reviewer's earlier concerns!
>
> We wish to express our sincere appreciation to the reviewer for recognizing the substantial significance of our Reti-Diff's contribution.
>
> Your acknowledgment holds great importance to us and serves as a meaningful validation of our dedicated efforts to advance this critical area of research.

---

### Author Response · Authors · 2024-11-19
**Summary of the rebuttal by authors**

We extend our sincere gratitude to all the reviewers (**R1**-**p8Wx**, **R2**-**uVsX**, **R3**-**NsRN**, and **R4**-**72tr**) for their insightful and considerate reviews, which help us to emphasize the contributions of our approach. We are pleased to hear that the reviewers approved the well-structured presentation of our research (**R3, R4**), the novelty of our work (**R2, R3, R4**), and its commendable performance (**R2, R3, R4**).

We are delighted to see reviewers confirm our contributions to the field of illumination degradation image restoration. These encompass our pioneering LDM-based framework, **Reti-Diff**, the plug-and-play Retinex prior estimator, **RLDM**, and the prior-guided image restoration network, **RGformer**.

In direct response to your thoughtful comments, we have methodically addressed each point in our individual responses, and we provide a summary here:

- We reorganized the paper to enhance clarity, highlighting the motivation and contributions while correcting writing errors.
- We added more qualitative and quantitative experiments, along with efficiency analyses, to further underscore our superior performance.
- We explored the potential of our approach under extremely degraded scenarios, demonstrating the generalizability of Reti-Diff.
- We provided additional ablation studies to validate the robustness and effectiveness of the various modules and strategies within Reti-Diff.

Once again, we deeply appreciate the reviewers' valuable suggestions. We have updated the paper accordingly and will release our code for community use and further study. We sincerely hope to **continue the discussion with the reviewers to ensure all concerns have been fully addressed**. If any aspects of our work remain unclear, we kindly invite further feedback to help us improve.

---

### Comment · Area_Chair_M2KB · 2024-11-20

Dear reviewers, do the authors' reply address your concerns? This is a reminder that November 13 to November 26 at 11:59pm AoE: Reviewers and authors can exchange responses with each other as often as they wish. Thanks!

---

### Meta-Review · Area_Chair_M2KB · 2024-12-21

**Metareview:**

(a) Summary:
The paper introduces Reti-Diff, a novel framework leveraging Retinex-based Latent Diffusion Models (RLDM) and Retinex-guided Transformers (RGformer) for illumination degradation image restoration (IDIR). The method estimates compact Retinex priors to enhance degraded images, achieving state-of-the-art results on multiple IDIR tasks and downstream applications like detection and segmentation, while maintaining computational efficiency.

(b) Strengths:
Innovative Approach: Combines Retinex theory, latent diffusion models, and transformers for robust illumination restoration.
Comprehensive Experiments: Validates effectiveness across multiple IDIR tasks and challenging conditions.
Efficiency: Achieves high performance with reduced computational cost.
Clear Presentation: Well-organized, easy to understand, and supported by accessible code.
(c) Weaknesses:
Reliance on Retinex Theory: Heavily depends on Retinex decomposition, with limited discussion on its reliability and robustness.
Gaps in Related Work: Lacks recent citations and explanation for design choices like cross-attention and training strategies.
(d) Recommendation:
The paper offers significant contributions to IDIR tasks, demonstrating innovation and strong empirical results. Despite minor concerns, it provides a solid foundation for future research.

Decision: Accept.

**Additional Comments On Reviewer Discussion:**

During the rebuttal period, the authors addressed all reviewer concerns with detailed responses and significant updates to the paper.

Reviewer p8Wx: Initially critical, the reviewer raised concerns about Retinex theory, novelty, and comparisons. After receiving detailed clarifications and additional experiments, they revised their score from negative to positive, indicating partial satisfaction.

Other Reviewers: The remaining three reviewers were highly satisfied with the authors’ rebuttals and appreciated the added clarity, experiments, and analyses.

Summary of Changes:
Improved paper organization and clarified motivation and contributions.
Added qualitative and quantitative experiments, including efficiency analyses, to support performance claims.
Demonstrated generalizability under extreme degradation scenarios.
Conducted new ablation studies to validate the robustness and effectiveness of individual modules.
These changes strengthened the paper, addressing all major concerns, and influenced the decision to accept, given its improved clarity, robust experimental validation, and demonstrated generalizability.

---

### Decision · Program_Chairs · 2025-01-22

Accept (Spotlight)